# Graphene Nanocomposites in the Targeting Tumor Microenvironment: Recent Advances in TME Reprogramming

**DOI:** 10.3390/ijms26104525

**Published:** 2025-05-09

**Authors:** Argiris Kolokithas-Ntoukas, Andreas Mouikis, Athina Angelopoulou

**Affiliations:** 1Department of Pharmacy, School of Health Sciences, University of Patras, GR-26504 Patras, Greece; kolokithas@upatras.gr (A.K.-N.); andreas27997@gmail.com (A.M.); 2Regional Centre of Advanced Technologies and Materials, Czech Advanced Technology and Research Institute (CATRIN), Palacky University Olomouc, 779 00 Olomouc, Czech Republic; 3Metabolic Engineering and Systems Biology Laboratory, Institute of Chemical Engineering Sciences, Foundation for Research and Technology-Hellas (FORTH/ICE-HT), GR-26504 Patras, Greece

**Keywords:** graphene, nanomaterials, graphene oxide, cancer, photothermal therapy, photodynamic therapy

## Abstract

Graphene-based materials (GBMs) have shown significant promise in cancer therapy due to their unique physicochemical properties, biocompatibility, and ease of functionalization. Their ability to target solid tumors, penetrate the tumor microenvironment (TME), and act as efficient drug delivery platforms highlights their potential in nanomedicine. However, the complex and dynamic nature of the TME, characterized by metabolic heterogeneity, immune suppression, and drug resistance, poses significant challenges to effective cancer treatment. GBMs offer innovative solutions by enhancing tumor targeting, facilitating deep tissue penetration, and modulating metabolic pathways that contribute to tumor progression and immune evasion. Their functionalization with targeting ligands and biocompatible polymers improves their biosafety and specificity, while their ability to modulate immune cell interactions within the TME presents new opportunities for immunotherapy. Given the role of metabolic reprogramming in tumor survival and resistance, GBMs could be further exploited in metabolism-targeted therapies by disrupting glycolysis, mitochondrial respiration, and lipid metabolism to counteract the immunosuppressive effects of the TME. This review focuses on discussing research studies that design GBM nanocomposites with enhanced biodegradability, minimized toxicity, and improved efficacy in delivering therapeutic agents with the intention to reprogram the TME for effective anticancer therapy. Additionally, exploring the potential of GBM nanocomposites in combination with immunotherapies and metabolism-targeted treatments could lead to more effective and personalized cancer therapies. By addressing these challenges, GBMs could play a pivotal role in overcoming current limitations in cancer treatment and advancing precision oncology.

## 1. Introduction

Graphene (Gr) and graphene-based materials (GBMs), such as graphene oxide (GO) and reduced graphene oxide (rGO), have attracted considerable interest in biomedical research, especially for their application in targeting solid tumors, owing to their unique physicochemical characteristics, high surface area, and ease of functionalization. Recent studies have further demonstrated the potential of GBM nanomaterials in cancer diagnosis and therapy, emphasizing their high drug-loading capacity, stable fluorescence imaging, enhanced photothermal effects, and cytotoxic activities against cancer cells [1,2]. Their application in cancer therapy and diagnostics (theranostics) stems from their unique physical, chemical, and biological properties [3]. GBMs, owing to their exceptionally high surface area, efficiently support the attachment of various therapeutic agents, including chemotherapeutic drugs [4], RNA molecules (like siRNA and mRNA) [4,5], or anticancer proteins and peptides [6], enhancing their effectiveness and potentially reducing side effects [7]. A focal point of GBM research has been the ease of their functionalization with various molecules and ligands to improve tumor-targeting specificity [5]. Their capacity to adsorb and deliver a broad spectrum of therapeutic agents has positioned GBMs at the forefront of cancer theranostics. However, while their versatility and multifunctionality have been well documented, their practical translation into clinical oncology remains constrained by several unresolved issues. Chief among these are their variable toxicity profiles. Particularly, the surface functionalization of GBMs has gained significant interest in enhancing or tailoring their biocompatible properties and optimizing their biosafety, since their toxicity profile may strongly vary depending on size, surface charge, and hydrophobicity [8,9]. To mitigate these limitations, extensive efforts have focused on the covalent and non-covalent modification of GBMs with biocompatible polymers and co-polymers, such as polyethylene glycol (PEG) [10], dextran [6], physical or hemisynthetic polysaccharides [11], and poly(lactic acid-co-ethylene glycol) (PLA-co-PEG) [12], has substantially improved their biological behavior in terms of solubility and colloidal stability [6], and improved the cellular accumulation and toxicity in tumor cells and normal cells [13]. The surface functionalization of GBMs for exploiting their effectiveness in biomedical applications has been an issue of interest in extensive reviews recently [1,2,3,4,5,6,7,8,9,10,11,12,13]. Especially, the surface modification with targeting ligands, such as antibodies, peptides, or small RNA molecules, can allow GBMs to increase their selectivity and binding affinity to tumor cell receptors that play an intimate role in carcinogenesis and tumor progression [4,5,14]. Such tumor-specific therapies improve the accumulation of therapeutic agents at the tumor site, while minimizing off-target side effects. While surface modification with tumor-targeting ligands (e.g., antibodies, peptides, small RNAs) enhances cellular specificity [4,5,14], there remains limited clarity on how these modifications influence long-term outcomes and resistance mechanisms in heterogeneous tumor microenvironments.

GBMs have emerged as key driving nanomaterials for overcoming tumor microenvironment (TME) biological barriers to promote deep tumor penetration and anticancer effectiveness (Figure 1). Their small size and ultrathin 2D plane provide them the ability to intercalate into tumor cell membranes and navigate through the TME extracellular matrix (ECM), enhancing their therapeutic efficacy [15]. Nevertheless, the clinical translation of this capability remains in early stages, partly due to the heterogeneity and adaptability of the TME itself. The TME is a highly differentiating network with altered metabolic status favoring immune reprogramming, thus contributing to tumor cell differentiation and immune cell suppression [16]. The protein-rich ECM of the TME consists of cell-secreted factors, including growth factors, cytokines, chemokines, and tumor antigens supporting cancer cell differentiation and mutation. In this ECM, cancer cells continuously escape immune surveillance, eventually leading to tumor growth and progression. The ECM, abnormal vasculature, and interstitial matrix represent the non-cellular components of tumor stroma that coexist alongside the cellular components that most commonly include the capillary and vascular cells, immune cells (innate and adaptive), mesenchymal stromal cells (MSCs), and cancer-associated fibroblasts (CAFs) [15,16,17]. While the literature robustly documents [15,16,17] the role of the TME in fostering immune evasion and therapeutic resistance, less attention has been given to the dynamic interplay between metabolic reprogramming and immune suppression. Recent findings underscore how the tumor stroma ECM acts as a steppingstone for the evolution of the metabolic alterations of tumor cells by supplying nutrients and growth factors, highly supporting the formation of metabolic networks between the stromal and the surrounding cancer cells [18]. Such metabolic networks offer support for the transfer of signaling molecules (metabolites) from stromal cells (immune and non-immune) to neighboring cancer cells, further promoting their proliferation, survival, and resistance. The stroma-derived metabolites serve as intermediates of paracrine signaling for the support of metabolite-responsive signaling pathways [19]. The metabolic support from stroma cells through paracrine signaling most commonly involves (i) CAFs that mainly secrete alanine and supply glutamine to promote cancer cell differentiation, (ii) peripheral axons that secrete serine to regulate mRNA translation, and (iii) macrophages that secrete pyrimidines, promoting drug resistance [18,19]. Moreover, through the activation of glycolysis and glutamine metabolism, tumor stroma cells (especially CAFs) manage to create a reservoir of abundant molecule supply to promote metabolic flexibility and allow cancer cells to adjust their metabolic phenotype adaptation in the TME [20]. This metabolic plasticity, in combination with the increased oncogenic signaling within the TME, supports tumor cell survival and drug resistance, limiting the efficacy of therapeutic agents (chemo- and immunotherapies). Yet, despite identifying key stromal contributors, such as CAFs, macrophages, and peripheral neurons, strategies to selectively disrupt these metabolic exchanges remain limited. GBMs, due to their modifiable surface chemistry and biological versatility, are uniquely positioned to disrupt these metabolic networks. However, there is a critical need for more targeted studies to evaluate how functionalized GBMs interact with specific metabolic circuits within the TME and whether they can modulate stromal-derived metabolic support without harming normal tissue homeostasis.

The cellular heterogeneity of the TME, particularly within the tumor stroma, significantly contributes to the metabolic diversity and reprogramming of both cancer and stromal cells. This metabolic remodeling leads to profound immune suppression within the TME [21]. A central driver of this process is the susceptibility of cancer and stromal cells to metabolic stress, which stems from their heightened energy and nutrient demands. The factors contributing to this metabolic stress mainly include the increased competition in areas with high metabolic activity and the limited oxygen and nutrient supply, owing to the abnormal vasculature [22]. Thus, cancer cells fuel their bioenergetic demands by alternative processes, such as aerobic glycolysis for the conversion of glucose to lactate, highly affecting the extracellular levels of both metabolites and resulting in the upregulation of glycolytic enzymes, pH gradient, and hypoxia signaling [23]. Moreover, cancer cells use glutaminolysis for glutamine conversion to glutamate that is further catalyzed to αKG (alpha-ketoglutarate) from glutamate dehydrogenase (GDH) to enable mitochondrial ATP production. In parallel, glutaminolysis becomes a key compensatory mechanism, as glutamine via the tricarboxylic acid (TCA) cycle provides carbon for the biosynthesis of lipids and metabolites and nitrogen for biosynthetic processes of amino acids and nucleotides, facilitating sustained proliferation [24]. Additionally, cancer cells exploit lipid metabolic pathways, engaging in both lipogenesis and lipolysis to meet energy demands and membrane biosynthesis needs. Through these processes, the de novo synthesis of monounsaturated fatty acids (FAs) and the lipoprotein lipase (LPL)-mediated extracellular lipolysis are promoted for additional supply with FAs, being essential building blocks of lipids and cholesterol synthesis [25,26]. While this metabolic flexibility is well characterized, it remains unclear how these processes differ between tumor types or across progression stages. Lipid metabolic reprogramming has been characterized as an essential cascade for cancer cells’ supply of FAs to support their increased growth and proliferation demands [26], and for stromal cells’ (especially immune cells) adaptability in the TME [25]. The immune cells (innate and adaptive) rely on the same bioenergetic resources for their function and intercellular and intracellular communications [27]. The metabolic heterogeneity and the extensive crosstalk among cancer and stromal cells determine the fate of nutrients and metabolites within the TME, shaping cancer cell progression and immune cell suppression. Emerging evidence also implicates stromal cells, such as CAFs and macrophages, in fueling cancer metabolism [28,29,30,31,32]. A distinct example is the increased production of reactive oxygen species (ROS) by the cancer cells that regulate the activation of CAFs, further stimulating aerobic glycolysis. CAFs, activated by ROS from tumor cells, increase glycolytic output and shuttle lactate via monocarboxylate transporter (MCT) MCT4 to cancer cells through MCT1 transporters, establishing a metabolic symbiosis that reinforces cancer cell survival and metabolic recycling [28]. This interplay not only promotes tumor growth but also enhances immune evasion. The competitiveness among cancer and immune cells for nutrient and metabolite supply, in combination with the excessive consumption by the cancer cells, regulates the suppression of immune cells’ function [29]. For instance, in a study by Chang et al. [30], it was presented that glucose competition in the TME of a mouse sarcoma model resulted in increased glucose consumption by the cancer cells and the metabolic restriction of tumor-infiltrating CD8+ T lymphocytes (TILs), downregulating T-cell responsiveness. Specifically, the mTOR activity, glycolytic capacity, and IFN-γ expression of T-cells were impaired, promoting tumor growth and progression. Similar patterns have been observed in a recent study by Kim et al. [31] where, in lung adenocarcinoma (ADC) tissues from patients, cancer cell glycolysis was associated with increased hexokinase-2 (HK2) gene expression that promoted an immunosuppressive and pro-tumorigenic state, downregulating CD8+ T-cell infiltration and increasing regulatory T-cell (Treg) infiltration. Fewer CD8+ T-cells and a low ratio of CD8+ T-cells to Tregs were associated with poor patient survival, signifying the importance of glycose metabolism in immune system suppression. Despite this growing body of knowledge, several questions remain. For instance, how can metabolic interventions selectively impair cancer metabolism without compromising the function of tumor-infiltrating lymphocytes? Additionally, Qian et al. [32] highlighted that tumor-associated macrophage (TAM) heterogeneity and M2 dynamic polarization states can further promote TME immunosuppressive effects and the immune evasion of cancer cells, supporting drug resistance and metastasis. In this metabolically heterogeneous and immunosuppressive TME, immune evasion and drug resistance highly influence the effectiveness of immunotherapies and targeted therapies. These gaps highlight the need for more precise metabolic profiling of the TME and tailored therapeutic strategies. GBMs represent a promising avenue in this respect. Their unique physicochemical properties allow for metabolic targeting of metabolic pathways, such as aerobic glycolysis (the Warburg effect), mitochondrial respiration, hypoxia and oxidative stress, drug delivery, and immunomodulation. However, translating these properties into effective metabolic therapies will require a more nuanced understanding of the TME’s metabolic networks and their influence on therapeutic resistance [33].

## 2. Graphene-Based Nanocomposites in the Reprogramming of the TME

The therapeutic potential of targeting the TME through nanomedicines has garnered substantial interest, particularly in addressing the metabolic reprogramming of cancer cells within the TME [17,34] in response to immune reprogramming [35], immunotherapies [36], the inflammatory microenvironment [37], and the microvasculature [38]. Within the TME, the cancer cells dispose of the efficiently healthy metabolic pathways and follow alternative routes that, through metabolic reprogramming, meet their biosynthesis and survival requirements [23]. The metabolic reprogramming of the TME affects the phenotype of cancer and stromal cells, including CAFs and immune cells, regulating cancer cell growth and immune cell suppression. The cellular crosstalk within the metabolic network of the TME is supported by signaling molecules and oncogenes, playing a vital role in cellular communications. Recently, there has been a great effort in targeting the metabolic pathways of the TME by exploiting the advantages of nanomedicine [33]. However, despite promising results, several challenges remain in fully exploiting these pathways for effective treatment. Several studies have shown promising results in exploiting TME vulnerabilities by targeting metabolic shifts, but inadequate specificity and the dynamic nature of metabolic reprogramming remain unresolved issues [23,33]. The difficulty in targeting TME metabolic pathways to impair and exhaust cancer cell bioenergetics with current nanomedicine approaches emerged from the complexity of the metabolic networks within the TME, encompassing the activity and regulation of multiple metabolites, enzymes, and proteins. Additionally, metabolic reprogramming of the TME can be highly affected by the communications among cancer and noncancerous cells (such as noncancerous epithelial cells and immune cells). The lack of specificity in metabolic targeting is highly affected by the metabolic flexibility of the TME, highlighting the necessity of tumor metabolome analysis in patients [39,40]. Further research is necessary to elucidate the roles of oncogenes and metabolic regulators in the TME, which could unlock novel therapeutic strategies for targeting metabolic networks more effectively.

The role of oncogenes in regulating cancer cell metabolism has been pivotal in advancing our understanding of metabolic reprogramming within the TME [40]. The role of oncogenes is of vital importance in the proliferation of cancer cells, stimulating processes that promote nutrient uptake, energy supply (via ATP biosynthesis), and the biosynthesis of proteins and lipids. However, while these insights are crucial, the specific mechanistic details of how oncogenes contribute to metabolic shifts in various cancer types remain insufficiently understood. In contrast to normal cells that are mainly based on mitochondrial oxidative phosphorylation (OXPHOS) for their energy supply during differentiation processes, cancer cells accumulate energy for their differentiation through aerobic glycolysis to avoid ROS formation (the Warburg effect). The switch to excessive glucose metabolism, accompanied by glutamine and glycine influx by cancer cells, has participated in the establishment of a hypoxic metabolic milieu [33,40,41]. The precise molecular triggers behind the switch from OXPHOS to glycolysis are still debated, presenting a gap in research that could lead to more targeted therapies. In particular, the role of metabolites such as glutamine and glycine in establishing the hypoxic microenvironment is a critical area where research is still catching up [33,40,41]. An understanding of the underlying mechanisms of cancer metabolism and the complex metabolic reprogramming is still to be determined; however, the metabolites, enzymes, proteins, and transporters that take part in metabolic pathways represent useful therapeutic targets for nanomedicine therapeutics. Despite these gaps, understanding the vulnerabilities of the TME metabolic milieu of cancer cells has led to the identification of potential therapeutic targets, including enzymes, metabolites, and transporters within the TME. Through metabolic reprogramming, it becomes possible to lead cancer cells to energy deprivation by restraining their biochemical resources and to enhance the effectiveness of anticancer therapies and immunotherapies [33]. Yet, while nanomedicine-based strategies (Figure 2) have demonstrated promise in modulating TME metabolism, the heterogeneity of tumors and the complexity of metabolic networks continue to pose challenges. Current nanotherapeutic approaches often struggle with selectivity and penetration into tumor tissues, limiting their efficacy. More research is needed to refine the delivery and functionalization of nanomaterials, ensuring they can effectively target oncogene-driven metabolic pathways without affecting normal tissues. The combination of metabolic inhibitors with nanomedicine, though revolutionary in theory, still faces significant hurdles in clinical application, suggesting that the future of this approach requires a more tailored and personalized therapeutic strategy to overcome these obstacles [42,43].

The physicochemical properties of GBMs extend beyond the densely packed pristine hexagonal lattice and the π-π stacking and hydrophobic interactions. The covalent and non-covalent surface functionalization of GBMs has been extensively explored to enhance their biocompatibility, bioavailability, and hydrophilicity, as well as to facilitate the transport of hydrophobic drugs, genes, and antibodies for targeted anticancer drug delivery [4,5]. Oxidized forms of graphene, such as GO, CGO, and rGO, have shown promising adhesion properties when interacting with extracellular matrix (ECM) proteins, genes, and growth factors within the TME [6,7]. This ability to enhance cell adhesion is orchestrated by electrostatic and hydrophobic interactions, accompanied by covalent and hydrogen bonding. However, while these interactions are well documented, there remain significant uncertainties about the long-term biocompatibility and toxicity of GBMs, especially in vivo, and their overall effectiveness in sustained drug delivery under physiological conditions. Furthermore, GBMs’ use in diagnostic biosensors and tumor imaging devices has garnered substantial attention, yet the clinical translation of these technologies remains hindered by challenges related to the scalability, cost, and precision targeting of biomarkers [44,45]. The integration of GBMs in photothermal (PTT) and photodynamic (PDT) combinational anticancer therapies is the main research interest of extended reviews [11]. GBMs, owing to their ability to absorb near-infrared radiation (NIR) light and transmit heat, manage to induce local heat generation (hyperthermia) and promote thermal ablation of cancer cells. Also, GBMs have been extensively studied for PDT, acting as photosensitizers due to their light absorption ability. Within the cellular TME, the GBMs transmit the absorbed energy-generating ROS (such as singlet oxygen, superoxide anion radicals, hydroxyl radicals, and hydrogen peroxide) at cytotoxic levels, promoting oxidative stress to cancer cells [46]. In the photothermal and photodynamic effects of GBMs, mitochondrial respiration, pH and hypoxic milieu, and the graphene-cell interactions highly affect ROS production [47]. However, while the mechanisms underlying these effects have been studied extensively, critical questions about the consistency of ROS production, the role of mitochondrial respiration, and how cellular pH and hypoxic conditions influence the efficacy of GBMs remain unanswered. Such interactions have also been crucial for the development of metabolic heterogeneity within the TME. Specifically, the interactions between GBMs and the complex cellular TME, including tumor cells, stromal cells, and immune components, are not yet fully understood, and the modulation of metabolic heterogeneity within the TME by GBMs remains underexplored. The lack of standardized protocols to evaluate these interactions and the variability in experimental conditions are major barriers to drawing clear conclusions regarding the effectiveness of GBMs in therapeutic applications. In a recent review, the ability of GBMs to induce and modulate autophagy and the molecular mechanisms of graphene-cell interactions were highlighted [42]. Autophagy is a critical process for cancer cells to promote their survival within the heterogenic TME to overcome nutrient and metabolite starvation by using autophagy-mediated recycling to maintain mitochondrial function and energy homeostasis [48,49]. Thus, autophagy inhibition by GBMs may be a beneficial target for the metabolic reprogramming of the TME. In a recent study by Zhang et al. [50], the effects of thermal ablation in plasma samples from patients with non-small cell lung cancer (NSCLC) revealed a significant impact on the regulation of metabolites associated with angiogenesis and inflammatory responses. CT-guided percutaneous microwave ablation was performed for patients after the administration of local anesthesia with a needle, reaching the tumor lesion, and with ablation being performed for 6–9 min at 40–60 W power, depending on the lesion size. Thermal ablation reduced the expression levels of pro-tumor-associated proteins (angiopoietin 1, ANGPT1, tyrosine kinase, and TIE2) and metabolic pathways by promoting a decreased ratio of metabolites (fatty acids, palmitic acid, and eicosapentaenoic acid) and steroid-related metabolites (corticosterone, cortisone, and cortisol) that participate in tumor angiogenesis and inflammatory responses. According to this study, after thermal ablation, pro-inflammatory proteins and metabolites induced elevated systemic inflammatory responses. Although there has been increasing interest in the ability of GBMs to induce autophagy, the exact pathways and regulatory networks through which GBMs influence autophagic processes are still poorly defined. More robust studies are required to better elucidate their mechanisms of action, their interaction dynamics within the complex TME, and their clinical applicability in combination with other cancer therapies.

Highly active metabolic pathways within the TME lead to critical alterations in the supply of nutrients and small molecules. One such pathway, aerobic glycolysis, is predominantly used by cancer cells for glucose metabolism in the mitochondria. The oxidation of glucose, fatty acids, and amino acids by mitochondria is promoted by the TCA cycle and the electron transport chain (ETC). This shift towards aerobic glycolysis, commonly referred to as the Warburg effect, allows cancer cells to meet their biosynthetic and energy demands. It is critical to note, however, that while aerobic glycolysis promotes rapid glucose metabolism, it also generates an excess of lactate and acidosis, which could limit long-term efficacy due to the impact of an acidic TME on therapy response. This way, excessive amounts of energy are produced for cancer cell proliferation and differentiation. Aerobic glycolysis is favored by cancer cells due to the rapid glucose metabolism accompanied by carbon and NAD^+^ (nicotinamide adenine dinucleotide) regeneration that is a critical metabolite for cellular homeostasis [51]. NAD^+^ is formed by the linkage of adenosine monophosphate (AMP) and nicotinamide mononucleotide (NMN). The phosphorylation of AMP results in the formation of NADP^+^ (nicotinamide adenine dinucleotide phosphate). In the nicotinamide group of NAD^+^ and NADP^+^, the carbon atom of the pyridine ring in the opposite position of the primary amide (meta position) can accept a hydride anion (H^+^, 2e^−^) and provide the reduced forms of NADH and NADPH. The reduced forms (NADH and NADPH) undergo oxidation through the ETC to promote the synthesis of ATP (adenosine triphosphate). Moreover, the intermediates of aerobic glycolysis (such as 3-Phosphoglycerate, glucose-6-phosphate, and fructose-6-phosphate) in combination with NAD^+^/NADP^+^ cofactor reduction are central to several biosynthetic pathways, including the pentose phosphate pathway (glucose-6-phosphate), serine synthesis (3-Phosphoglycerate), and fatty acid synthesis (pyruvate) [52]. While these pathways are crucial for cell growth, a critical gap in the current research is understanding how the interplay between these metabolites, NAD^+^/NADP^+^ reduction, and TME signaling contributes to cancer cell adaptation and treatment resistance. Another key metabolic pathway regulated by mitochondria is the hexosamine biosynthesis pathway (HBP), which is vital for the synthesis of nucleotide sugars that play a fundamental role in the glycosylation of proteins and lipids. The flux through this pathway, which is tightly regulated by glucose-6-phosphate and fructose-6-phosphate, also depends on glutamine as an amide donor. Glutamine, a major nitrogen reservoir, plays a vital role in amino acid and nucleic acid biosynthesis, underscoring the tight coupling between glucose and glutamine metabolism in cancer cells [53]. Thus, glucose and glutamine metabolism are highly associated with cancer cell proliferation. The TCA cycle and glutamine oxidation also contribute to key metabolites, such as aspartate and malate, which are essential for NADPH production and redox homeostasis in cancer cells. Apart from the TCA cycle, ETC is essential to ATP production and mitochondrial aspartate formation by the catabolism of glutamine in cancer cells. Aspartate and malate are the two major metabolites deriving from mitochondrial glutamine oxidation within the Krebs cycle (KC) functioning in normal cells. These metabolites are used for NADPH production that is crucial in anabolic processes and redox homeostasis. For aspartate production, glutamine enters the KC as α-ketoglutarate (α-KG), to be enzymatically converted to glutamate by glutaminase (GA). Glutamate is then further converted to α-KG by glutamate dehydrogenase (GLUD1), glutamate-oxaloacetate transaminase (GOT2), and glutamate pyruvate transaminase (GPT2), further releasing ammonia, aspartate, and alanine, respectively. While significant research has elucidated how glutamine enters the TCA cycle as α-ketoglutarate (α-KG) and is subsequently converted to glutamate, there remains a knowledge gap regarding the precise role of mitochondrial glutamine metabolism in maintaining cellular bioenergetics and tumorigenesis.

In cancer cell metabolism, GPT1 (cytosolic) and GPT2 (mitochondrial) are essential for energy metabolism by providing alanine for protein biosynthesis and feeding TCA cycle intermediates. Moreover, GOT acts as a pool by providing cytosolic aspartate for protein and nucleotide synthesis. Furthermore, glutamine serves as a precursor for the de novo enzymatic synthesis of asparagine from aspartate, which is catalyzed by asparagine synthetase (ASNS). This process is closely linked to mTORC1 activation (mammalian target of rapamycin mTOR complex 1), a key regulator of cell growth and protein synthesis, further contributing to the synthesis of purines and pyrimidines [53]. The emerging link between metabolic reprogramming and immune suppression suggests that metabolic inhibitors targeting glucose and glutamine metabolism may hold promise for enhancing immunotherapy efficacy [54]. The dysregulation of aerobic glycolysis and HBP pathways in cancer cells has a great impact on the immune cells’ signaling, cytokine production, and oncogene regulation, thus influencing immune suppression. To facilitate TME metabolic reprogramming, tumor metabolism has been encountered as a checkpoint promoting immune suppression. In this aspect, metabolic inhibitors have been studied for targeting tumor metabolism by the blockade of glucose and glutamine metabolism for cancer cell energy exhaustion [51,52,53,54].

However, challenges remain in targeting these pathways without inducing significant toxicity to normal cells, highlighting the need for refined therapeutic strategies to selectively disrupt tumor metabolism while preserving immune function. In conclusion, while targeting the metabolic pathways within the TME presents promising therapeutic opportunities, critical questions remain regarding the mechanisms of action, the crosstalk between cancer and immune cells, and the efficacy of metabolic inhibitors in clinical settings. The dysregulation of key metabolic pathways has profound implications for immune suppression and tumor progression, but much more work is required to fully understand the complexity of these pathways and their potential as therapeutic targets.

### 2.1. Graphene and Graphene Quantum Dots

The generation of mitochondrial ROS by the activation of metabolic pathways (TCA cycle and ETC) in cancer cells plays an essential part in tumor progression and metastasis. The elevated ROS formation is regulated by the cytosolic antioxidant system, which includes enzymatic antioxidants such as SOD (superoxide dismutase), CAT (catalase), GPX (glutathione peroxidase), TXN-TXNRD (thioredoxin—thioredoxin reductase), and non-enzymatic antioxidants, such as GSH (glutathione), vitamins, selenium, and metabolites (bilirubin, melatonin). The enzymatic and non-enzymatic antioxidants create an antioxidant network that regulates the expression of critical oncogenes (mutant KRAS and MYC genes), leading to the suppression of the antioxidant defense mechanism. In cancer progression, the presence of increased ROS expression levels can promote genomic and metabolic instability by the inactivation of tumor suppressor genes and mitogenic signaling pathways. However, a critical threshold exists in cancer cells being regulated by the imbalance between ROS production and cellular antioxidants present in the cytosol. This imbalance may cause increased oxidative stress, damaging proteins, lipids, and DNA, eventually leading to oxidative cancer cell death [55,56]. Thus, while endogenous antioxidant networks regulate basal ROS levels, many cancer cells exhibit elevated oxidative stress due to metabolic reprogramming. This paradoxically promotes both tumor growth and vulnerability to ROS-targeting strategies. Graphene-based nanomaterials, particularly graphene quantum dots (GQDs), have emerged as promising tools for manipulating mitochondrial redox balance. The dysfunction of mitochondrial ROS formation is a critical determinant of cancer therapeutics, being targeted by disrupting the OXPHOS mechanism, by scavenging the ROS production, and by promoting oxidative stress cell death (Figure 3).

Among GBMs, graphene (Gr) and graphene quantum dots (GQDs) have been investigated for ROS-targeting due to their superior photothermal/photodynamic light conversion ability after UV/Vis and NIR light radiation. Especially, GQDs possess unique physicochemical properties—such as tunable photoluminescence, excellent mitochondrial penetration due to their small size (<10 nm), and facile surface functionalization—that make them ideal for ROS modulation. Functionalized GQDs have been shown to accumulate mitochondria, destabilize cellular membranes, and enhance oxidative stress under photothermal/photodynamic therapy (PTT/PDT) conditions. For instance, Perini et al. [57] demonstrated that the functionalization of GQDs had a great impact on membranes’ fluidity upon their interaction with U87MG glioblastoma cancer cells and glioblastoma neurospheres. The carboxylated GQDs significantly altered membrane fluidity and impaired glioblastoma neurosphere formation. The surface charge-driven membrane destabilization not only disrupted cell integrity but also sensitized tumor cells to subsequent therapeutic interventions. These findings suggest GQDs can interfere with lipid metabolism, a key driver of tumor plasticity and invasiveness. Membrane fluidity is a critical aspect for cancer cell progression and metastasis, since the altered lipid metabolism promotes lipid-protein heterogeneities in the cell membrane composition. Particularly interesting is the expression of phosphatidylcholine that regulates the expression levels of enzymes and energy metabolites, promoting cancer cell progression. Moreover, the composition of the cellular membrane in highly metastatic cancer cells is associated with reduced cholesterol content, thus increasing their plasticity and fluidity to promote blood vessel penetration [58]. Table 1 outlines the characteristic research studies of Gr and GQDs nanocomposites presented herein for the reprogramming of TME.

Using a synergistic approach, Perini et al. [59] explored the co-administration of carboxylic-acid-functionalized GQDs with doxorubicin (DOX) and temozolomide (TMZ) for the combined PTT effect on a 3D spheroid model of glioblastoma. While TMZ (an alkylating agent) remains a cornerstone of glioblastoma treatment, its efficacy is limited by the tumor’s redox-adaptive mechanisms. Glioblastoma cells are highly dependent on redox regulation, often exploiting antioxidant systems to resist ROS-induced damage; thus, elevated intracellular ROS production promotes TMZ resistance through impaired redox balance [60]. The upregulation of ROS levels was studied as an effective way to target redox homeostasis to induce oxidative stress that may further promote cell death and sensitize cancer cells to anticancer therapies [60]. In this aspect, Perini et al. [59] demonstrated that the synergistic effect of PTT photosensitizing therapy highly promoted DOX membrane permeability, further enhancing intracellular accumulation and efficacy. Notably, this increased intracellular uptake in 3D spheroid models under NIR irradiation led to heightened therapeutic response, highlighting the potential of GQDs not merely as passive carriers, but as active modulators of drug delivery and redox homeostasis. Yet, while the enhanced membrane disruption is promising, it raises concerns about off-target effects in non-tumor tissues. Moreover, the lack of long-term in vivo data on biodistribution and tumor selectivity limits the immediate translational value of these findings. A more detailed mechanistic investigation into how GQDs alter membrane composition and lipid raft dynamics could further clarify their selectivity. Moreover, the elevated presence of tumor-associated antigens and ROS (due to GQDs-induced PTT) promoted the migration of immune cells at the tumor site, suggesting the reactivation of immune responses. Thus, beyond drug delivery, the immunomodulatory role of GQDs is particularly noteworthy. In this study, increased tumor-associated antigen expression and immune cell infiltration post PTT were observed, suggesting that GQDs may also contribute to tumor immunogenicity. However, whether this effect leads to durable anti-tumor immune responses or immune exhaustion remains to be clarified in future immunocompetent models.

Oxidative stress targeting has also been studied recently by Campbell et al. [61] in multifunctional nitrogen-doped GQD formulations functionalized with hyaluronic acid (HA) and ferrocene (Fc) (Fc-GQDs-HA) and designed for dual PTT fluorescence imaging of HeLa cervical cancer cells. This composite leverages several tumor-targeting strategies: HA for CD44 receptor targeting, Fc for Fenton-mediated redox toxicity, and N-GQDs for optical imaging and ROS induction. The N-doped GQDs (N-GQDs) are pronounced fluorophores for cellular and tumor multicolor imaging. Moreover, N-GQDs were applied in the targeting of the acidic extracellular TME, since they presented a pH-dependent fluorescence response [62]. The functionalization of N-GQDs with HA served in the targeting of the CD44 cancer cell receptor overexpressed in HeLa cells and played a key part in the regulation of cell proliferation and survival. The conjugation of Fc facilitated a redox-based toxicity in the cancer cells throughout the redox cycle of iron. Specifically, the anticancer properties of Fc and its derivatives (photochemical and biochemical organometallic compounds) originate from the Fenton pathway, which promotes the oxidation of Fe(II) to Fe(III), leading to increased ROS production [63]. The central iron atom of Fc presents great redox capacities through its electron donor-acceptor ability, catalyzing the Fenton reaction of endogenous superoxide molecules and promoting the oxidation of Fc to ferricinium cation. The Fe(III) ions are used in lipid metabolism, promoting the generation of increased levels of lipid hydroperoxides that further induce ferroptosis (iron-dependent cell death) in cancer cells [64,65]. This multifunctionality is significant for enhancing tumor specificity. In this study, the HA moiety enabled the selective accumulation in CD44-overexpressing cancer cells, offering a promising strategy to reduce off-target toxicity. Simultaneously, the Fc moiety catalyzed intracellular ROS generation through Fenton chemistry, promoting oxidative damage and potential ferroptosis. The redox cycling of Fc (Fe^2^⁺/Fe^3^⁺) played a dual role—stimulating lipid peroxidation and triggering iron-dependent cell death pathways. However, while the theoretical mechanism of ferroptosis induction is compelling, the experimental evidence remains mostly indirect. Most studies rely on lipid peroxidation and membrane potential markers, but do not assess ferroptosis-specific biomarkers (e.g., GPX4 inhibition, ACSL4 expression). Moreover, the long-term stability of Fc within biological systems and its potential for off-target ROS production in healthy tissues are areas of concern requiring further investigation [64,65]. From an imaging standpoint, the N-doped GQDs exhibit strong fluorescence in acidic conditions, a property exploited here. This pH-dependent optical response could aid in theranostics; however, current studies lack quantitative analysis comparing tumor and normal tissue imaging contrast. Optimizing this optical selectivity remains a crucial step for clinical translation.

N-doped GQDs were also studied by Guo et al. [66] in mitochondria targeting the photo-responsive platform for the combined delivery of nitric oxide (NO) and triphenylphosphonium (TPP) for in vivo PTT anticancer therapy. The N-GQDs were surface functionalized with ruthenium nitrosyl, which served as the NO donor under NIR light irradiation. This system was designed for dual NO release and PTT therapy under NIR irradiation, aiming to exploit mitochondrial dysfunction as a therapeutic vulnerability in cancer. The essential role of NO in mitochondrial respiration is based on its ability to regulate oxygen consumption and ATP synthesis by serving as a redox-signaling molecule and promoting an inhibitory effect in the electron transport chain (ETC) of the mitochondria inner membrane. ETC is a collection of protein complexes (complex I, complex II, coenzyme Q, complex III, cytochrome C, and complex IV) that are used in the transportation (pass-through) of electrons in a series of redox reactions that release energy. The released energy further creates a proton gradient used in the ATP synthesis. NO mainly inhibits the action of complex IV (Cytochrome c oxidase (COX)), which is the terminal enzyme of ETC responsible for reducing molecular oxygen to H_2_O and driving ATP synthesis [67]. More details on the mechanisms of the NO inhibitory effect on the mitochondrial respiratory chain and biogenesis are presented in a review by Tengan et al. [68]. Defective mitochondrial respiration can be promoted in cancer cells by the prolonged exposure to increased levels of NO, further regulating elevated ROS and RNS (reactive nitrogen species) production. The ruthenium nitrosyl N-GQDs presented increased mitochondrial targeting due to the surface functionalization with TPP^+^ cations that effectively target the negatively charged mitochondrial membrane. TPP^+^ belongs to the family of delocalized lipophilic cations (DLCs) that are selectively transported through the mitochondrial phospholipid membrane. The hyperpolarized potential of the mitochondrial membrane can significantly be targeted by DLCs that, due to their lipophilicity, can be easily transported through the outer and inner mitochondrial membranes [69]. Thus, the therapeutic rationale by Guo et al. [66] was well designed, since NO is a known redox-signaling molecule capable of inhibiting ETC, thereby disrupting ATP synthesis, and the TPP functionalization enhances mitochondrial specificity by leveraging the organelle’s negative membrane potential for subcellular targeting. The mitochondria localized the elevated release of NO (due to ruthenium nitrosyl) under NIR light radiation in combination with the PTT effect, which significantly inhibited in vivo tumor growth. The targeting of mitochondrial membrane potential has been an ideal tool for increasing the selectivity and therapeutic efficacy of ROS-inducing GBMs. Importantly, the authors demonstrated in vivo tumor growth inhibition, highlighting the system’s therapeutic promise. Yet, several critical questions remain regarding the safety of heavy-metal-based NO donors like ruthenium complexes, particularly for systemic applications [68]. While the authors reported biocompatibility in preclinical models, long-term toxicological data and clearance pathways are still lacking. Moreover, although the platform induced significant mitochondrial depolarization and oxidative stress, it is unclear whether the dominant mechanism of cell death was apoptosis, ferroptosis, or necrosis [67]. Discriminating between these pathways is crucial for predicting resistance mechanisms and designing combinatorial strategies. In a study by Fan et al. [70], GQDs functionalized with TPP were studied for their application in the in vitro monitoring of mitochondria to enable precise mitochondrial localization. The GQDs were surface functionalized with positively charged PEI (polyethyleneimine), offering an amide linkage to TPP. The dual-functionalized GQDs-PEI-TPP graphene construct demonstrated efficient mitochondria targeting, as confirmed by the comparative fluorescence imaging of GQDs (green fluorescence) and mitochondria (Mitotracker Red fluorescence). While the imaging performance and subcellular localization are compelling, the study primarily served as a proof-of-concept for diagnostic tracking rather than therapeutic application. Building upon the concept of mitochondrial dysfunction as a therapeutic trigger, Zhang et al. [71] presented a more advanced platform of hybrid GQDs covalently functionalized with rare-earth-doped UCNP (upconversion nanoparticles). The GQDs-UCNP showed increased mitochondria specificity due to the conjugation of TRITC (Tetramethylrhodamine-5-isothiocyanate). Isothiocyanate (ITC) molecules are promising anticancer agents that express elevated mitochondria targeting, leading to ITC-induced apoptotic cell death by activating the caspase-9-related apoptotic pathway and by regulating Bcl-2 protein family members related to pro-apoptotic signaling in cancer cells. For the activation of caspase-9, mitochondria release cytochrome c from their intermembrane space, inducing the loss of mitochondrial membrane integrity [72,73]. The key innovation lies in the mitochondrial specificity conferred by TRITC, a molecule with known mitochondrial affinity and apoptosis-inducing properties. The TRITC-functionalized GQDs-UCNP promoted the increased in situ cytotoxic ROS formation (mainly singlet oxygen species, ^1^O_2_) in mitochondria upon NIR light irradiation. Moreover, the mitochondrial dysfunction was confirmed by the significant decrease in mitochondrial membrane potential and the activation of the caspase 3 apoptotic pathway. However, while these data support apoptosis induction, the potential overlap with the ferroptosis or necroptosis pathways should be explored further, particularly given the increased ROS levels and mitochondrial membrane disruption. Notably, the in vivo results in 4T1 tumor models showed that mitochondrial damage is the main signal promoting apoptotic cell death in cancer cells; thus, the combined mitochondrial targeting and PDT effect resulted in important tumor suppression upon NIR irradiation in 4T1 tumor models treated with GQDs-UCNP/TRITC.

Graphene was also applied in mitochondria-targeting anticancer therapy by Saeed et al. [74] in breast and pancreatic cancer cells. Graphene was used for the nanodelivery of gambogic acid (GA), significantly enhancing its in vitro bioavailability. GA is a natural caged xanthonoid compound that highly promotes calcium ion (Ca^2+^) intracellular influx, efficiently promoting the dysregulation of Ca^2+^ homeostasis in various cancer cells. The increased upregulation of Ca^2+^ promotes the dysfunction of the mitochondrial membrane and the subsequent activation of mitochondrial apoptosis signaling [75]. The mitochondrial damage is promoted by GA through the disruption of lipid metabolism, with the swelling of mitochondria (mega-mitochondria) and the endoplasmic reticulum (ER vacuolization) leading to paraptosis (a distinct form of programmed cell death associated with vacuolization) [76,77]. The graphene-GA highly promoted the depletion of mitochondrial membrane potential (MMP) and decreased the intracellular level of lipid droplets serving in energy storage. Moreover, nuclear DNA fragmentation was induced by the graphene-GA, proving apoptotic cancer cell death. However, GA was not the only compound contributing to the disruption of MMP. In a study by Li et al. [78], commercial pristine graphene was able to be internalized into murine RAW 264.7 macrophages and promote mitochondrial pathway signaling to induce apoptotic cell death. Specifically, graphene triggered increased ROS levels and activated pro-apoptotic proteins of the Bcl-2 family (Bim and Bax) through the activation of MAPK and TGF-β signaling that stimulated the activation of the caspase-3-related apoptotic pathway. Thus, pristine graphene was able to induce cytotoxicity to murine macrophage-like cells by regulating the mitochondria-induced apoptotic cell death. Despite promising preclinical data, several limitations remain. Most studies, including those cited here, are conducted in vitro or in 3D spheroid models, with limited in vivo validation. Moreover, the specificity of GQDs for cancer cells versus healthy cells remains insufficiently characterized. The long-term biosafety, biodistribution, and potential off-target effects—especially in mitochondrial pathways crucial for normal cell function—are underexplored for both pristine and functionalized graphene. A more nuanced understanding of how graphene-induced ROS generation differentially impacts cancer versus healthy cells is crucial before advancing toward clinical applications. Additionally, while studies often emphasize enhanced ROS production, the precise thresholds between therapeutic oxidative stress and toxicity-induced side effects are rarely quantified. More comparative studies are needed to assess whether GQDs and Gr offer significant advantages over traditional photosensitizers in terms of efficacy, selectivity, and immune modulation.

**Table 1 ijms-26-04525-t001:** Applications of graphene (Gr) and graphene quantum dots (GQDs)-based nanocomposites in cancer therapeutics.

Carrier Type	Agent	Characteristics	Ref
GQDs Carboxylated-GQDs	-	increased membrane fluidity reduced formation of neurospheres in U87MG glioblastoma cancer cells and glioblastoma neurospheres	[57]
Carboxylated-GQDs	DOX, TMZ	synergistic PTT on 3D spheroid model of glioblastoma, elevated intracellular ROS production, increased membrane permeability, elevated presence of tumor-associated antigens	[59]
nitrogen-doped GQD with HA and Fc	HA, Fc	CD44 cancer cell receptor targeting in HeLa cells, Fc facilitated a redox-based toxicity by the redox cycle of iron, oxidative stress targeting	[61]
nitrogen-doped GQD	TPP, ruthenium nitrosyl	mitochondria targeting, PTT, regulation of oxygen consumption and ATP synthesis, an inhibitory effect in the ETC, inhibited the in vivo tumor growth	[66]
GQDs-PEI	TPP	in vitro mitochondria monitoring	[70]
Hybrid GQDs-UCNP	TRITC	promoted increased in situ cytotoxic ROS formation, mitochondria dysfunction, decrease in mitochondria membrane potential, activation of caspase-3 apoptotic pathway	[71]
Graphene	GA	promoted the depletion of MMP, decreased the intracellular level of lipid droplets, induced DNA fragmentation	[74]
Pristine graphene	-	promote apoptotic mitochondria pathway signaling in murine RAW 264.7 macrophages, triggered increased ROS levels, activated Bcl-2 family pro-apoptotic proteins, activated MAPK and TGF-β signaling, activated caspace-3-related apoptotic pathway	[78]

Abbreviations: PTT: photothermal therapy, DOX: doxorubicin, TMZ: temozolomide, ROS: reactive oxygen species, HA: hyaluronic acid, Fc: ferrocene, TPP: triphenylphosphonium, ETC: electron transport chain, UCNP: upconversion nanoparticles, TRITC: Tetramethylrhodamine-5-isothiocyanate, GA: gambogic acid, MMP: mitochondria membrane potential, PEI: polyethyleneimine.

Figure 4 presents characteristic research studies of Gr and GQDs nanocomposites in TME reprogramming.

### 2.2. Graphene Oxide

Hypoxia, acidity, oncogenic mutations, and immunosuppression constitute a TME that hinders the efficacy of anticancer therapies and immunotherapies by accumulating toxic metabolites and promoting nutrient starvation that induces an aggressive cancer cell phenotype and mechanisms of multidrug resistance (MDR). In response, emerging therapeutic strategies rely on the versatility and facility of GBM functionalization, especially GO, to deliver targeted nanomedicines for the reprogramming and modulation of TME. GO’s therapeutic potential stems from its abundant binding oxygen functional groups, along with its intrinsic hydrophobicity and π-π stacking interactions of its pristine graphene lattice. Nano-GO has presented significant cellular transportation, either by endocytosis and plasma membrane mechanisms (intracellular) or by diffusion (paracellular pathways). Through functionalization (Figure 5), GO enabled the binding of targeting ligands for cancer cell membrane receptors and mitochondrial membrane to selectively deliver anticancer and immunotherapeutic agents for TME reprogramming.

In a recent study by Han et al. [79], tumor and mitochondria targeting were assessed by the conjugation of hypericin (HY) on PEG-functionalized GO through disulfide bonding. HY is a natural pleiotropic product that acts as an anticancer agent by inhibiting pro-inflammatory mediators and increasing the expression levels of caspase-3 and caspase-4, promoting apoptotic cell death [80]. In the acidic and hypoxic TME, HY selectively binds to the low-density lipoprotein (LDL) receptors of cancer cell membranes, promoting its non-specific accumulation [81,82]. Mitochondria are the primary target of the photodynamic action of HY by producing singlet oxygen molecules (^1^O_2_) through a type II oxygen-dependent photosensitization mechanism in the ETC at complex III [83]. Then, the generation of free radicals and reactive intermediates is promoted, leading to the induction of mitochondrial membrane permeability transition [84]. This chain of events illustrates how GO can amplify the mitochondrial-mediated apoptosis pathway under physiologically relevant conditions. The HY-functionalized GO vehicle was studied for the combined PDT and anticancer therapy of breast cancer in vitro and in vivo by the synergistic action of doxorubicin (DOX). The GO vehicle was internalized via endocytosis, promoting the glutathione-triggered release of HY and the pH-triggered release of DOX. Upon NIR laser irradiation, HY targeted the mitochondria, generating singlet oxygen that further promoted the release of cytochrome c into the cytosol. The DOX-induced DNA damage and simultaneous cytochrome c release further promoted caspase activation that stimulated apoptotic cancer cell death. Together, these stimuli demonstrate a multi-pronged strategy for enhanced apoptosis. Similarly, Wu et al. [85] extended this approach by engineering a multifunctional GO nanocomposite with DSPE-PEG2000 as a nanographene platform for the combined PDT/PTT and immunostimulatory anticancer effect in vitro and in vivo in the EMT-6 transplantable mouse mammary tumor cell line. The amphiphilic DSPE-PEG was attached to the graphitic lattice of GO via hydrophobic interactions. Mitochondria targeting was demonstrated by the GO nanocomposite through the conjugation of the lipophilic cation alkyl-triphenylphosphonium (TPP^+^) in the DSPE-PEG chain. The alkylated TPP^+^ has been extensively used in mitotargeting due to the negative membrane potential of the mitochondrial inner membrane, resulting in the increased uptake of positively charged small molecules [86]. The PDT/PTT effect of GO was boosted by the IR820 NIR-responsive photosensitizer to promote ROS-induced mitochondrial damage due to NIR-light conversion. Furthermore, the GO nanocomposite exhibited significant immunostimulatory effects owing to the linkage of CpG ODN nucleotide acid drugs (synthetic unmethylated cytosine-guanosine oligodeoxynucleotides). The CpG ODN are short, single-stranded synthetic DNA molecules that contain unmethylated oligodeoxynucleotides. CpG ODNs are recognized by TLR9 (toll-like receptor 9), highly expressed on human B cells and pDCs (plasmacytoid dendritic cells), to stimulate immune responses (innate and adaptive) [87]. The combinational effect of CpG ODN with PDT/PTT resulted in increased ROS production that promoted the mitochondrial-induced cell death and the significant upregulation of proinflammatory cytokines (such as IL-6, TNF-α, and INF-γ), improving immunogenicity in the TME and inhibiting tumor growth by 88%. Taken together, these studies underscore several key trends: the integration of multiple stimuli-responsive mechanisms (e.g., pH, GSH, NIR light), the convergence of cytotoxic and immunostimulatory effects, and the increasing emphasis on mitochondrial targeting as a vulnerability in cancer cells. However, while preclinical efficacy is promising, the complexity of these nanoplatforms may pose translational challenges in terms of large-scale production, pharmacokinetics, and off-target effects. Addressing these will be critical for the clinical success of GO-based TME therapies. Table 2 outlines the research examples of GO nanocomposites in TME reprogramming that are presented here.

The synergistic effect of mitochondria-targeting and PDT/PTT has emerged as a promising approach to overcome drug resistance in cancer. Zeng et al. [88] explored this synergistic potential against doxorubicin-resistant osteosarcoma (OS) by designing a multifunctional GO nanocomposite. The system involved polyethylenimine (PEI)-modified PEGylated GO nanosheets functionalized with indocyanine green (ICG) and the mitochondria-targeting ligand (4-carboxybutyl) triphenyl phosphonium bromide (TPPB). ICG is an FDA-approved NIR fluorescent contrast agent for imaging and image-guided surgery. Owing to its ability to convert the absorbed light to energy, ICG has been used in cancer theranostic applications as a photosensitizer to induce hyperthermia and increase ROS expression levels as singlet oxygen (^1^O_2_) and superoxide (O^2−^) [89]. TPPB facilitated the specific accumulation of the nanocomposite in mitochondria, while ICG enabled NIR-triggered ROS generation and hyperthermia. Upon NIR irradiation, the increased intracellular localization of the nanocomposites enabled the synergistic PDT/PTT therapy of doxorubicin-resistant MG63/DOX OS cancer cells in vitro and in vivo in OS tumor-bearing mice. The elevated production of intracellular ROS (^1^O_2_, O^2−^), due to ICG, in combination with the GO hyperthermia effect, resulted in the irreversible apoptotic damage of cancer cells through mitochondrial dysfunction. This damage disrupted ATP synthesis, depriving drug-resistant cancer cells of their energy supply, thereby enhancing apoptotic cell death. The mito-damage was associated with the inhibition of ATP synthesis that further suspended the energy supply of MG63/DOX cancer cells. The suppressed ATP production in vivo upon NIR light resulted in the inhibition of tumor growth and the increased presence of apoptotic or necrotic cancer cells in tumor tissues. The GO nanocomposites expressed elevated biosafety as it was assessed by blood biochemistry markers (alanine aminotransferase ALT, aspartate aminotransferase AST, and alkaline phosphatase ALP), indicating minimal systemic toxicity. Complementary findings were reported by Wei et al. [90], who developed a GO nanocomposite modified with an integrin αvβ3 monoclonal antibody (mAb) and a pyropheophorbide-a (PPa) photosensitizer that was conjugated with PEG polymer, evaluated as on/off phototoxicity switches for tumor and mitochondria-targeted delivery. The switch on/off mechanism was based on the hydrophobic interactions between PPa and the graphitic plane of GO in an aqueous environment, such as blood circulation and the cytoplasm of cancer cells. Within an aqueous environment, the design leveraged the hydrophobic and π–π stacking interactions between PPa and GO to create a “phototoxicity-off” state in aqueous environments, such as the bloodstream or cytoplasm. This structural stability was maintained until laser irradiation activated a fluorescence resonance energy transfer (FRET) mechanism, allowing energy transfer from PPa to GO and initially quenching phototoxicity. The GO nanocomposites exhibited a preferential accumulation into the mitochondria, achieved through αvβ3 mAb targeting, which facilitated internalization and efficient in vitro delivery to mitochondria in U87-MG glioblastoma and MCF-7 breast cancer cells. The system provided an effective distribution of PPa and translocation from the lysosome to the mitochondria. The underlying mechanism for effective mito-translocation was associated with the electrostatic interaction between the negatively charged mitochondrial membrane and the polarized GO nanocomposites that promoted their physical attraction to the mitochondria. Once localized in the mitochondria’s lipid-rich environment, the π-π stacking interactions among PPa and GO were destabilized, triggering a conformational switch to a “phototoxicity-on” state. Upon light irradiation, the switched-on state of the PPa photosensitizer permitted the localized ROS formation into the mitochondria, initiating apoptosis via mitochondrial membrane disruption. These two studies underscore the growing trend toward stimuli-responsive, mitochondria-specific nanotherapeutics that integrate diagnostic and therapeutic functionalities. Both platforms highlight how careful material design, targeting ligands, FRET mechanisms, and redox/light sensitivity can fine-tune therapeutic precision while mitigating systemic side effects. Nevertheless, such complexity, while powerful, raises translational challenges related to manufacturing consistency, regulatory approval, and cost-effectiveness. These hurdles must be addressed to realize the clinical potential of GO-based combinational therapies.

The targeting of mitochondria is an effective approach to reprogram TME-promoting apoptotic cancer cell death. Beyond conventional targeting ligands, mitochondria-targeting peptides (MitPs) have also shown promise in facilitating the intracellular delivery of therapeutics. Zhu et al. [91] studied the in vitro application of magnetic GO nanocomposites conjugated with MitP for the selective delivery of mitoxantrone (MTX) and the promotion of mitochondrial dysfunction. MTX is a DNA topoisomerase II inhibitor that forms a DNA-topoisomerase II/MTX complex to inhibit DNA transcription and replication processes [92]. However, its clinical use is limited by cardiotoxic side effects, which stem from MTX-induced mitochondrial dysfunction, including membrane potential collapse, ATP depletion, and mitochondrial swelling [93]. The MitP-GO nanocomposites presented increased localization in the mitochondria of A549 lung adenocarcinoma cancer cells, promoting the selective delivery of MTX. Upon exposure to an alternating magnetic field (AMF), which enhances cellular uptake and localized heating, the system promoted a downregulation in the expression levels of ATP and a decrease in mitochondrial membrane potential, which present hallmarks of mitochondrial dysfunction. This disruption was further confirmed by the release of cytochrome c and the subsequent activation of the caspase-3 apoptotic pathway. A related approach was developed by Zhang et al. [94], who designed a multifunctional β-cyclodextrin (β-CD)-grafted GO nanoassembly incorporating TPM-Azobenzene (TPM-Azo). The TPM-Azo was prepared by covalent functionalization of polylysine (Plys) with the tumor-targeting protein transferrin (TF), the MitP, and PEG to improve biodispersibility and biocompatibility. The inclusion of azobenzene (Azo), a hypoxia-activated molecule, added a further layer of control. Azo is a photoactive molecule activated in the absence of oxygen that can act as a nitrogen mustard deactivator, thereby facilitating selective mitochondrial targeting while avoiding off-target toxicity, particularly in the liver. Nitrogen mustards can generate intra- and inter-strand crosslinking with DNA, inducing DNA damage. However, the DNA repair mechanisms of cancer cells may promote drug resistance and mutagenic effects. The reapplication of Azo into the mitochondria is based on the presence of mitochondria-targeting ligands or peptides to increase mito-localization, bypass cellular drug resistance mechanisms, and off-target liver activity [95]. The MitP promoted the in vitro accumulation of the GO nanoassemblies in A549 lung cancer cells, further inducing mitochondrial aggregation and decreasing ATP expression levels. The application of UV and NIR irradiation promoted the dissociation of the TMP-Azo from the GO nanoassemblies, further stimulating mitochondrial aggregation and decreasing ATP levels. This mitochondrial dysfunction was associated with the disruption of the cancer cell cycle, inducing arrest at the G2 phase, thus preventing cancer cells from entering mitosis and inhibiting cancer cell proliferation. Thus, the G2 cell cycle arrest was related to decreased A549 cell viability. Under NIR-induced photothermal activation, the PTT transition activity of the nanoassemblies significantly decreased cell viability and increased cytochrome c expression levels from the mitochondria into the cytosol. The in vivo application in S180 tumor-bearing mice demonstrated significant tumor growth inhibition, indicating the therapeutic potential of these multifunctional GO-based nanoassemblies. Collectively, these studies illustrate the evolving sophistication of GO-based nanocarriers, incorporating peptide targeting, light-responsive components, and multifunctional therapeutic agents. By leveraging mitochondrial vulnerability in cancer cells, these systems offer highly specific, multi-modal strategies to overcome drug resistance and reshape the hostile conditions of the TME. Future studies should prioritize understanding long-term safety, biodegradability, and off-target effects to support clinical translation.

Mitochondria-triggered apoptosis remains a cornerstone mechanism for effective anticancer therapy, particularly when combined with targeted drug delivery platforms. Zhang et al. [96] investigated glycyrrhetinic acid (GA)-functionalized GO nanocomposites for the selective delivery of doxorubicin in mitochondria. GA is a biologically active metabolite that derives from glycyrrhizin, a natural product of Glycyrrhiza glabra (or licorice) with broad anticancer properties and mitochondrial targeting ability [97]. The anticancer properties of GA are related to the inhibition of protein kinase C (PKC) α/βII and the activation of c-Jun NH_2_-terminal kinase of the JNK pathway, promoting the apoptosis of non-small cell lung cancer cells [98]. In the mitochondria, GA can significantly decrease mitochondrial membrane potential and promote caspase-3-mediated apoptosis [99] and restrict energy metabolism by binding to and inhibiting the activity of the enzyme serine hydroxymethyltransferase 2 (SHMT2) [97]. GA, by interacting with complex I of the mitochondrial respiratory chain, promotes oxidative stress, facilitating the generation of ROS, such as hydrogen peroxide (H_2_O_2_), superoxide radicals, and the highly reactive hydroxyl radicals that further lead to thiol oxidation. This mechanism promotes the opening of the mitochondrial permeability transition pores in the presence of calcium ions (Ca^2+^), inducing mitochondria-targeting effects on GA [100]. The GA-functionalized GO system enabled pH-responsive DOX release and enhanced mitochondrial accumulation in HepG2 hepatocellular carcinoma cells. The synergistic effect of GA triggered mitochondrial membrane permeabilization by inducing mitochondrial dysfunction [100]. The opening of the mitochondrial permeability transition pores was verified by the increase in the ratio of Bax/Bcl-2 proteins, facilitating the release of cytochrome c into the cytosol and further activating the caspase-mediated apoptotic pathway (especially caspase-3). These mitochondrial events were validated both in vitro and in vivo in hepatocellular carcinoma-bearing nude mice, demonstrating potent anticancer efficacy. In a parallel study, Jiang et al. [101] developed GO nanodrugs functionalized with GE11 EGFR targeting peptide for the delivery of oridonin (Ori). Ori is a natural terpenoid interfering in the apoptotic signaling pathways by modulating the Bcl-2/Bax protein expression. The Bcl-2 and Bcl-xl belong to the anti-apoptotic protein family, while the Bax and Bak belong to the pro-apoptotic protein family. Thus, the downregulation of Bcl-2, in combination with upregulation of Bax expression, promoted the release of cytochrome c to the cytoplasm, leading to caspase activation and caspase-triggered apoptosis [102]. The GO-Ori nanodrug was evaluated in vitro in KYSE-30 and EC-109 human esophageal squamous cell carcinoma cells that overexpress the EGFR receptor. GO-Ori nanodrug effectively promoted mitochondrial dysfunction by decreasing the mitochondrial membrane potential and increasing the Bax/Bcl-2 ratio. Moreover, the increased lysosomal accumulation of the GO-Ori nanodrug significantly promoted toxicity and induced cell cycle arrest, increasing the G2/M phase. The induced apoptosis was thus dual-faceted: both mitochondria-mediated and cell cycle dependent. Importantly, the EGFR-targeted delivery of Ori led to the downregulation of the Ras/Raf/MEK/ERK signaling axis—a critical pathway involved in cancer cell proliferation, migration, and survival. Together, these findings reinforce the value of functionalized GO nanoplatforms for multi-pathway targeted therapy, combining mitochondrial stress, apoptotic pathway activation, and receptor-specific delivery for improved anticancer outcomes.

Graphene oxide (GO) has emerged as a promising nanomaterial for targeting the TME, particularly due to its unique interactions with cancer stem cells (CSCs), which play a pivotal role in tumor progression, metastasis, and resistance to therapy. The physicochemical properties of GO, such as its large surface area and functional groups, enable it to selectively interact with CSCs via receptor-mediated endocytosis or by modulating signaling pathways crucial for CSC survival and self-renewal [103,104,105]. A study by Fiorillo et al. [106] provided important insight into the potential of graphene oxide (GO) to modulate cancer stem cell (CSC) behavior, specifically by inducing their differentiation and impairing tumor-sphere formation. The proposed mechanism involves the inhibition of key signaling pathways known to sustain CSC stemness, including Wnt, Notch, and STAT3, as well as suppression of the NRF2-dependent antioxidant response. Notably, the authors observed that the GO flakes used (5–20 μm) exceeded the size of CSCs, suggesting that internalization was unlikely and that the observed effects were likely mediated at the cell surface, where several of these signaling pathways are initiated. This surface-level interaction is particularly relevant for Notch and STAT3, which are known to regulate self-renewal and survival. However, a critical limitation is the non-specificity of the STAT3 pathway, the inhibition of which could negatively affect normal stem cells as well. This underscores the importance of distinguishing between CSC-specific and general stem cell signaling during therapeutic development. While the findings are promising, especially in the context of aggressive tumors such as glioblastoma (GBM), further investigation is required to validate these effects in vivo and to elucidate the precise molecular interactions between GO- and CSC-related signaling networks. Such studies would be instrumental in assessing the feasibility and safety of GO-based therapies targeting CSC populations. Building upon Fiorillo et al.’s [106] findings regarding the surface-level interaction of GO with CSC-related signaling pathways, a complementary approach involves the functionalization of GO as a drug carrier via mesenchymal stem cells (MSCs). Recent studies by Suryaprakash et al. [107] have demonstrated that GO can be efficiently loaded with chemotherapeutics such as doxorubicin and mitoxantrone and subsequently anchored to the surface of MSCs without compromising their viability or tumor-tropic behavior. This dual-component delivery system capitalizes on GO’s large surface area and pH-responsive drug release, along with MSCs’ intrinsic ability to home to tumor sites. Notably, MSCs coated with GO-drug complexes selectively induced cytotoxicity in glioblastoma and breast cancer cells while remaining viable themselves, suggesting a promising therapeutic index. The system also retained its migration capacity and showed enhanced CXCR4 expression (C-X-C chemokine receptor type 4 or CD184), supporting the chemotactic targeting mechanism. However, the burst release profile within 72 h may pose limitations in targeting dispersed tumor cells or CSCs located in more resistant niches. Moreover, the use of large GO flakes, though advantageous for surface interaction, may hinder cellular uptake and long-term biocompatibility. While these results support the feasibility of MSC-GO systems as versatile and targeted drug carriers, in vivo validation and a deeper understanding of the interplay between GO, drug release kinetics, and CSC subpopulations are required to advance this platform toward clinical translation. Building on previous strategies, the study by Kang et al. [108] introduces an innovative cell-mediated delivery system using AuNP/GO hybrid sheets anchored to MSC surfaces. This design circumvents key limitations of intracellular nanoparticle loading, such as cytotoxicity and exocytosis, by stabilizing the therapeutic cargo externally. The use of α-synuclein-coated AuNPs ensures tight packing on both sides of GO flakes, enhancing plasmon coupling and thereby amplifying photothermal effects upon NIR irradiation. Importantly, MSCs retained their tumor-homing capacity despite surface modification, achieving greater nanoparticle accumulation in tumor sites. The reported increase in photothermal ablation efficacy, compared to free or internalized AuNPs, underscores the potential of this platform. Nonetheless, questions remain about the long-term biodistribution and immune recognition of MSC-cloaked nanomaterials in vivo. Future work should assess repeated dosing, clearance, and potential off-target heating. Still, this work highlights the power of combining nanomaterial engineering with cell-based delivery for precision oncology. It complements previous findings on GO-CSC interactions and MSC-based systems, paving the way for multifunctional nanocarriers in targeted therapy.

**Table 2 ijms-26-04525-t002:** Applications for graphene oxide (GO)-based nanocomposites in cancer therapeutics.

Carrier Type	Agent	Characteristics	Ref
GO-PEG	HY, DOΧ	Combined PDT and anticancer therapy, internalization via endocytosis, glutathione-triggered HY release, pH-triggered DOX release, HY-triggered generation of singlet oxygen in mitochondria, cytochrome c release into the cytosol, caspases activation	[79]
GO-DSPE-PEG_2000_	TPP^+^, IR820 NIR Photosensitizer, CpG ODN	combined PDT/PTT and immunostimulatory anticancer effect, mitochondrial targeting, increased ROS production, mitochondrial induced cell death, upregulation of proinflammatory cytokines (IL-6, TNF-α, INF-γ)	[85]
GO-ICG	TPPB	increased mitochondrial accumulation, synergistic PDT/PTT, elevated ROS (^1^O_2_, O^2−^) production, GO hyperthermia effect, inhibition of ATP synthesis	[88]
GO-PEG	integrin α_v_β_3_ mAb, PPa	on/off phototoxicity switches, FRET mechanism, mitochondria accumulation, localized ROS formation, promoting cell apoptosis	[90]
GO-MitP	MTX	Increased mitochondria localization, promoting mitochondria dysfunction upon AFM, decrease in MMP, downregulation in ATP expression levels, cytochrome c release, activation of caspase 3 apoptotic pathway	[91]
GO-β-CD/Plys/PEG/MitP	TPM-Azo, TF	Increased mitochondria accumulation, stimulating mitochondrial aggregation, reducing ATP levels, disruption of cancer cells cycle, arrest at the G2 phase, decreased cell viability, increased cytochrome c expression into the cytosol	[94]
GO-GA	DOX	mitochondria targeting, selective pH-dependent DOX release, decreased the MMP, opening of the mitochondrial permeability transition pores, increase in the ratio of Bax/Bcl-2 proteins, activation of caspase-mediated apoptotic pathway	[96]
GO-GE11	Ori	promoted mitochondrial dysfunction, decreased the MMP, induced cell cycle arrest increasing the G2/M phase, downregulation of the Ras/Raf/MEK/ERK pathways	[101]
GO-CSCs	-	surface-level interaction, the inhibition of key signaling pathways Wnt, Notch, and STAT3, suppression of the NRF2-dependent antioxidant response	[106]
GO-MSCs	DOX, MTX	pH-responsive drug release, enhanced CXCR4 expression, induced cytotoxicity in glioblastoma and breast cancer cells	[107]
AuNP/GO-MSCs	-	amplified photothermal effects upon NIR, increase in photothermal ablation efficacy, tumor-homing capacity	[108]

Abbreviations: PEG: polyethylene glycol, HY: hypericin, PDT: photodynamic therapy, DOX, doxorubicin, DSPE-PEG_2000_: 1,2-distearoyl-sn-glycero-3-phosphoethanolamine-poly(ethylene glycol)-2000, PTT: photothermal therapy, TPP^+^: alkyl-triphenylphosphonium, NIR: near-infrared region, CpG ODN: synthetic unmethylated cytosine-guanosine oligodeoxynucleotides, ROS: reactive oxygen species, ICG: indocyanine green, TPPB: (4-carboxybutyl) triphenyl phosphonium bromide, mAb: monoclonal antibody, PPa: pyropheophorbide-a photosensitizer, FRET: Fluorescence Resonance Energy Transfer, MitP: mitochondria targeting peptide, MTX: mitoxantrone, AFM: alternating magnetic field, MMP: mitochondrial membrane potential, ATP: Adenosine triphosphate, β-CD: β-cyclodextrin, TPM-Azo: TPM-Azobenzene, Plys: polylysine, TF: transferrin, GA: glycyrrhetinic acid, GE11: EGFR-targeting peptide, Ori: Oridonin, CSCs: Cancer Stem Cells, MSCs: Mesenchymal Stem Cells, CXCR4: C-X-C chemokine receptor type 4 or CD184, AuNP: Gold nanoparticles.

GO has emerged as a promising material for targeting the TME, offering advantages such as high surface area, ease of functionalization, and low toxicity when tailored appropriately. Its ability to interact with CSCs and modulate key signaling pathways is a potential tool for overcoming tumor recurrence and drug resistance. Additionally, GO’s versatility in drug delivery platforms enhances therapeutic targeting and selective elimination of cancer cells. However, the limitations of GO, including concerns about its long-term biocompatibility, cellular uptake efficiency, and potential off-target effects, warrant caution. Moreover, the burst release kinetics of GO-loaded therapeutics may reduce its effectiveness in treating dispersed, invasive tumor cells. Despite these challenges, GO remains a promising candidate, and further research is needed to optimize its therapeutic application and ensure clinical safety.

Figure 6 visualizes the characteristic research studies of GO nanocomposites in the reprogramming of the TME.

### 2.3. Reduced Graphene Oxide

Given the centrality of mitochondrial dysfunction in tumor metabolic reprogramming to induce mitochondria-mediated apoptosis and energy deprivation, rGO has emerged as a promising nanomaterial for disrupting mitochondrial activity and triggering apoptosis in cancer cells. The ROS formation plays a crucial role in the regulation of mitochondrial function and reprogramming, with elevated ROS levels acting as signaling molecules that modulate cellular pathways involved in cancer cell survival and proliferation. ROS formation plays a dual role in the mitochondria of cancer cells, since it contributes to cellular adaptation, survival, and progression while also driving DNA damage and genomic instability, creating a mutagenic environment that drives the evolution of aggressiveness and metastasis [109]. Targeting the mechanisms that control ROS generation in mitochondria holds therapeutic promise in cancer treatment (Figure 7). The driving force behind using rGO for such applications stems from several of its unique physicochemical properties, which make it highly suitable for modulating ROS production and improving cancer treatment. Compared to GO, rGO has a lower ratio of oxygen groups, rendering it less hydrophilic but significantly more conductive. This higher electrical conductivity, combined with its small lateral dimensions and sharp edges, facilitates membrane penetration, intracellular uptake, and enhanced cytotoxicity [110]. As summarized in Table 3, several studies have demonstrated the potential of rGO-based systems in mitochondrial modulation and ROS-associated apoptosis.

In a study by Kretowski et al. [111], rGO induced autophagy and cell cycle arrest in MDA-MB-231 and ZR-75-1 breast cancer cells in vitro. This mitochondrial disruption was evidenced by a significant reduction in mitochondrial membrane potential and increased expression of cytosolic caspase-9 and caspase-3, suggesting activation of the intrinsic apoptotic pathway. Similarly, Zhang et al. [112] explored the synergistic effects of rGO and gamma-irradiation in H9C2 rat myocardial cells and C57BL/6 male mice. Their findings demonstrated that rGO could promote oxidative stress damage with elevated levels of ROS expression while reducing the mitochondrial membrane potential. Notably, in combination with radiation, rGO further impaired myocardial function by reducing serum levels of key metabolic enzymes such as aspartate aminotransferase and lactate dehydrogenase, indicating systemic toxicity and cumulative mitochondrial dysfunction. Table 3 presents research examples of rGO nanocomposites.

Certainly, one of the major advantages of rGO is its ability to generate ROS, especially singlet oxygen and free radicals upon light activation (NIR or UV-vis light), which is particularly useful in PDT [47]. When rGO or its functionalized derivatives are directed to mitochondria, ROS accumulation in these organelles can lead to oxidative damage, mitochondrial dysfunction, and apoptotic cell death. rGO can directly interact with mitochondria directly, either by localizing on their surface or internalizing into the organelle, facilitating ROS-induced damage. The effect of rGO in the production of ROS can be greatly improved by its surface functionalization to provide engineered rGO, enabling better targeting of cancer cells and mitochondria with targeting molecules like peptides or antibodies that bind specifically to cancer cell markers or mitochondria-associated receptors [110]. In a recent study by Vinothini et al. [113], rGO was co-functionalized with 4-hydroxycoumarin (4-HC) photosensitizer and magnetic nanoparticles (MNPs), forming a multifunctional nanocarrier for camptothecin (CMP) delivery under UV light-mediated PDT. The synergistic chemophotodynamic effect of the rGO nanodrug highly promoted cytotoxicity of MCF-7 human breast cancer cells through elevated production of ROS and regulation of apoptotic cell death. Apoptosis was characterized by typical hallmarks, including cell shrinkage and nuclear fragmentation, as well as upregulation of pro-apoptotic p53 and Bax proteins. The intracellular ROS accumulation peaked at 24 h post-irradiation and drove cell membrane damage via redox imbalance. The rGO nanodrug exhibited a synergistic antitumor effect in vivo in DMBA-induced mammary tumor-bearing rats, with decreased tumor volume/weight due to the increased apoptotic cell death and tumor tissue damage. These effects were mechanistically supported by increased expression of the p53 gene (a crucial biomarker of apoptosis and DNA damage) and Bax protein (regulation of apoptotic signaling pathways), coupled with a marked downregulation in enzymatic markers of mitochondrial function (acid phosphatase, β-D-glucuronidase, cathepsin, and LDH).

Beyond light-activated cytotoxicity, rGO has also shown promise in redox-sensitive therapies through synergistic interaction with enzymatic inhibitors that regulate proteostasis and cell cycle checkpoints. In a separate study, Kretowski et al. [114] investigated the combinatorial effect of rGO and the proteasome inhibitor MG-132 on apoptosis induction via oxidative stress. MG-132 was mixed with rGO to evaluate the anticancer effect of the nanocomposite against ZR-75-1 and MDA-MB-231 breast cancer cells. MG-132 is a peptide aldehyde that selectively inhibits the proteasome, which is a large multi-subunit protein complex present in the nucleus and cytoplasm of cells with the main function to degrade damaged, misfolded, or superfluous proteins by proteolysis. To be recognized, the proteins are tagged with ubiquitin (a small protein) and translocated into the catalytic core of the proteasome, allowing them to be degraded into smaller peptides. This process is essential for maintaining protein homeostasis (proteostasis) and regulating various cellular processes, such as cell cycle progression, signal transduction, and stress responses [115]. The MG-132 inhibitor can selectively induce apoptosis in glioblastoma cells by inhibiting the PI3K/Akt and NFκB signaling pathways involved in cancer initiation and progression. By promoting cell cycle arrest at the G2/M phase, the MG-132 prevented cancer cells from entering mitosis, thus inhibiting their proliferation. Moreover, the MG-132 inhibitor promoted mitochondrial dysfunction by decreasing the mitochondrial membrane potential and opening the mitochondrial permeability transition pore, resulting in the downregulation of the anti-apoptotic bcl-xl protein expression levels and further activation of caspase-3 apoptotic signaling [116]. When combined with rGO, MG-132 synergistically enhanced mitochondrial dysfunction by reducing membrane potential, promoting the opening of the mitochondrial permeability transition pore, and downregulating anti-apoptotic Bcl-xL expression. This cascade led to the activation of both caspase-3 and initiator caspases (caspase-8 and caspase-9), signifying dual-pathway apoptosis. Furthermore, the nanocomposite elevated intracellular oxidative stress by boosting ROS levels while concurrently depleting antioxidant defenses, including glutathione (GSH), superoxide dismutase (SOD), and glutathione peroxidase (GPx), thereby creating a redox-imbalanced environment that reinforced cell death mechanisms.

The importance of rGO nanocomposites with silver nanoparticles (AgNPs) in anticancer therapy has attracted attention specifically regarding ROS formation by mitochondria due to their ability to target and disrupt mitochondrial function, a key pathway for cancer cell apoptosis. The rGO/Ag nanocomposites significantly increase oxidative stress due to their ability to be internalized by cells, allowing them to interact directly with the mitochondria, increasing ROS levels in a more localized manner [117]. This synergistic ROS production is further enhanced by AgNPs, which can catalyze Fenton-like reactions by interacting with endogenous hydrogen peroxide (H_2_O_2_) to generate highly toxic hydroxyl radicals, thereby amplifying oxidative stress and mitochondrial damage in cancer cells. As an efficient electron donor, rGO facilitates redox reactions by enhancing electron transfer during redox reactions, leading to a higher concentration of ROS in the mitochondria. The combination with AgNPs can act synergistically to catalyze ROS production, significantly increasing ROS accumulation, leading to oxidative stress and further promoting mitochondrial dysfunction [118]. The anticancer properties of rGO-AgNPs nanocomposites were evaluated on their effect on TME by Gurunathan et al. [119] in A2780 epithelial ovarian carcinoma cells. The rGO-AgNPs nanocomposites expressed an increased inhibitory effect on cell viability accompanied by a pronounced decrease in cell growth. Moreover, the rGO-AgNPs significantly induced intracellular LDH leakage (lactate dehydrogenase) as a response to the damage of cellular membrane integrity. ROS generation was significantly promoted, inducing oxidative stress and increasing the intracellular expression level of pro-oxidant MDA (malonaldehyde) while decreasing antioxidant GSH (glutathione) production. The strong effect of rGO-AgNPs nanocomposites on inducing oxidative stress highly promoted caspase-3 activation and induced apoptotic cell death. The synergistic therapeutic effects between rGO-AgNPs highlight the modulation of oxidative stress and mitochondrial dysfunction as a central therapeutic mechanism, aligning with the focus on ROS-mediated anticancer strategies.

Yuan et al. [120] investigated the combination of rGO-AgNPs nanocomposites with cisplatin (Cis) to evaluate their in vitro synergistic effect on apoptosis and autophagy of HeLa human cervical cancer cells. The synergy of Cis with rGO-AgNPs highly inhibited the viability and proliferation of cervical cancer cells, promoting increased LDH leakage due to cellular membrane damage, in comparison to Cis, pristine rGO, and AgNPs. The rGO-AgNPs acted synergistically with Cis in supporting the oxidation-reduction status (redox) of cancer cells, promoting a pronounced increase in the Cis-induced ROS expression levels. The increased oxidative stress of the cancer cells was verified by the elevated MDA expression levels. Moreover, this combination had a strong influence on the loss of mitochondrial membrane potential and on gene expression levels related to mitochondrial dysfunction. This oxidative stress cascade was accompanied by significant transcriptional changes, including the upregulation of apoptotic genes (P53, P21, BAX, BAK, CASP3, and CASP9) and downregulation of anti-apoptotic genes (BCL2 and BCL2L1), resulting in DNA damage and the activation of apoptosis. The activation of CASP3 and CASP9 gene expression was associated with mitochondria-induced apoptosis pathway. Cisplatin exerts its cytotoxic effects primarily through DNA damage, leading to the activation of signaling pathways, including pathways for the regulation of autophagy. By disrupting DNA replication and transcription, Cis promotes the activation of DNA damage cellular responses, including the upregulation of p53 protein expression that is related to the promotion of BECN1 (Beclin-1) autophagy-related gene. Cis-induced DNA damage promotes oxidative stress and mitochondrial dysfunction, leading to a decrease in cellular ATP levels and AMPK protein kinase activation. In turn, the activation of AMPK can inhibit the mTOR pathway, promoting the induction of autophagy in response to Cis-induced stress. Collectively, DNA damage, oxidative stress, p53 activation, and AMPK/mTOR inhibition contribute to the cellular responses to Cis that can influence the activation of autophagy [121]. Interestingly, the combinational treatment of Cis with rGO-AgNPs significantly increased the induced autophagic cell death through the generation of autophagosomes and upregulation of autophagy-related (ATG) gene expression. The pronounced presence of autophagic vacuoles was combined with increased generation of autophagolysosomes. Under the elevated ROS production and oxidative stress promoted by the Cis and rGO-AgNPs combination, the extended presence of autophagy endorsed cancer cell death. These findings suggest that rGO-AgNPs could serve as potent sensitizers in combination therapies, enhancing the efficacy of conventional chemotherapeutics by amplifying mitochondrial and autophagy-related pathways. Moreover, this study links the nanocomposite’s action to both mitochondria-mediated apoptosis and autophagy-related pathways, showing dual cytotoxic modes of action. In short, the rGO-AgNPs underscore the therapeutic versatility and potency of rGO-based nanocomposites in a multimodal anticancer context, enhancing their overall value in the reprogramming of TME.

The ability of rGO to adsorb surface proteins plays a pivotal role in shaping its toxicological profile and cellular interactions. In MDA-MB-231 mammary epithelial cancer cells, it was shown that lower surfactant concentrations led to increased protein corona formation, which in turn enhanced ROS production, lipid peroxidation, oxidative stress, and inhibited mitochondrial ATP synthesis [122]. Surface functionalization of rGO has emerged as a strategy to modulate its interaction with cellular and mitochondrial membranes. In this context, Sawosz et al. [123] evaluated the functionalization of rGO with the amino acids arginine (Arg) and proline (Pro) for their anticancer effect in glioblastoma multiforme U87 cells in vitro and in tumors in vivo. These amino acids were conjugated onto rGO using a reduction process to improve stability and protect rGO from agglomeration. Both Pro and Arg play important roles in cellular metabolism, signaling, and stress responses. Proline metabolism and degradation pathways participate in regulating the function of p53 protein inducing stress signal responses, including DNA damage, oxidative stress, and nutrient deprivation. Such signals act through the mitochondrial function by proline dehydrogenase (PRODH) enzymatic action that converts proline to glutamate while generating ROS byproducts and activating p53 expression through oxidative stress mechanisms [124]. Arginine metabolism is important in cancer biology, as it can influence cell survival and apoptosis by regulating NO (nitric oxide) production through the action of nitric oxide synthase (NOS), further activating p53 expression. NO can activate the mitochondrial apoptotic pathway, leading to mitochondrial dysfunction and the release of pro-apoptotic factors like cytochrome c, which subsequently activates p53-mediated cell death pathways [125]. Functionalization of rGO with these amino acids enhanced its membrane adhesion and cytotoxicity. The resulting nanocomposites reduced the tumor volume by inhibiting cell proliferation that was accompanied by inhibition of FGF2 (fibroblast growth factor-2) and downregulation of VEGF expression. Apoptotic signaling was further supported by the activation of caspase-3 and the upregulation of MDM2 and COX6 expressions. Notably, Arg-functionalized rGO showed a stronger pro-apoptotic effect through activation of intrinsic apoptotic pathways and COX6 expression, while Pro-functionalized rGO appeared to mitigate these effects.

**Table 3 ijms-26-04525-t003:** Applications for reduced graphene oxide (rGO)-based nanocomposites in cancer therapeutics.

Carrier Type	Agent	Characteristics	Ref
rGO	-	stimulated autophagy and cell cycle arrest in cancer cells, promoted apoptosis signaling pathway, decreased the MMP, activated caspase-9 and caspase-3 cytosolic expression	[111]
rGO	Gamma irradiation	oxidative stress damage, elevated levels of ROS expression, damage of myocardial tissue, reduced expression levels of enzymes	[112]
rGO-MNPs	4-HC, CMP	Combined chemophotodynamic effect, elevated ROS production, regulation of apoptotic cell death, cell shrinkage and nuclear fragmentation, increased expression of p53 and Bax proteins	[113]
rGO	MG-132	increased apoptosis and necrosis of breast cancer cells, increasing ROS formation, caspase-8 and caspase-9 apoptotic pathways activation, reduced enzymatic antioxidants activity (SOD, GPx)	[114]
rGO-AgNPs	-	Decreased viability and cell growth, leakage of intracellular LDH, damage of cellular membrane integrity, ROS generation, increased MDA pro-oxidant levels, decreased antioxidant GSH production, activation of caspase-3 apoptotic death	[119]
rGO-AgNPs	Cis	Inhibition of viability and proliferation, increased Cis-induced ROS expression levels, increased oxidative stress, elevated MDA expression levels, loss of MMP, upregulation of apoptotic genes, downregulation of anti-apoptotic genes	[120]
rGO-Arg/Pro	-	increased cellular membranes adhesion, reduced cell proliferation, downregulation of VEGF expression, caspsase-3 activation, inhibition of FGF2, regulation of apoptosis	[123]

Abbreviations: MMP: mitochondrial membrane potential, ROS: reactive oxygen species, MNPs: magnetic nanoparticles, 4-HC: 4-hydroxycoumarin, CMP: camptothecin, MG-132: aldehyde proteasome inhibitor peptide, SOD: superoxide dismutase, GPx: glutathione peroxidase, AgNPs: silver nanoparticles, LDH: lactate dehydrogenase, MDA: malonaldehyde, GSH: glutathione, Cis: cisplatin, Arg: Arginine, Pro: Proline, FGF2: fibroblast growth factor-2.

In conclusion, while recent advancements in rGO-based nanocomposites, particularly those functionalized with silver nanoparticles and amino acids, have demonstrated significant promise in cancer therapy through the induction of ROS-mediated mitochondrial dysfunction and apoptosis, there remain key gaps in understanding and application. Notably, while the combination of rGO with chemotherapeutic agents such as cisplatin and mitoxantrone has shown synergistic effects, the underlying mechanisms, including long-term stability, toxicity, and immune response, have not been sufficiently explored. Furthermore, much of the current research is limited by reliance on in vitro models that do not accurately reflect the complexity of the tumor microenvironment or human physiology. Future research should focus on optimizing surface functionalization for better targeting, conducting in vivo studies to assess the translational potential, and exploring the combination of rGO nanocomposites with other treatment modalities to overcome the limitations of current therapies. This critical evaluation of the existing studies highlights both the potential and the challenges in advancing rGO-based cancer therapies. Importantly, the versatility of rGO-based systems allows for tailored functionalization, enhancing mitochondrial targeting and therapeutic efficacy while opening avenues for multimodal cancer treatment strategies such as chemo-, photo-, and redox-based therapies. This positions rGO as a promising nanomaterial for advancing the reprogramming of TME.

Figure 8 illustrates the characteristic research studies that were presented for the application of rGO nanocomposites in the reprogramming of TME.

## 3. Discussion

Reprogramming the highly heterogenic TME is a complex and challenging task, primarily due to its inherent diversity and dynamic nature. The TME is composed of various cell types, including cancer cells, stromal cells, immune cells, endothelial cells, and extracellular matrix components, all of which can vary significantly between different regions within the same tumor. This variability contributes to the immunosuppressive environment that hampers effective immune responses and therapeutic efficacy. Immune cells, such as regulatory T cells (Tregs), myeloid-derived suppressor cells (MDSCs), and tumor-associated macrophages (TAMs), dominate the TME, creating a microenvironment that impedes the immune system from efficiently targeting and eliminating tumor cells. Although the use of GBM in nanomedicines offers a promising strategy for reprogramming the TME for both immune-based and chemo-based nanomedicine therapies, this approach is not without its limitations. While GBM nanocomposites have demonstrated efficacy in overcoming the challenges of heterogeneity and immunosuppression, their ability to address the full spectrum of TME diversity remains an ongoing challenge [126]. The advancements in personalized nanomedicine, which aim to deliver drugs and therapeutic agents bypassing the physical barriers of the TME, represent a significant step forward. Personalized nanomedicine has become a promising approach for enhancing the precision and effectiveness of cancer therapies since they can combine multiple therapeutic modalities, including chemotherapy, radiotherapy, gene therapy, and immunotherapy, into a single nanoparticle platform. This multi-functional approach can overcome resistance mechanisms within the TME and achieve more comprehensive treatment outcomes [127]. However, questions about the stability, biocompatibility, and long-term safety of these nanomaterials persist, and it remains unclear whether these multi-functional approaches can maintain the desired therapeutic outcomes across the broad range of tumor types and stages. Moreover, the combination of advanced nanomaterials with GBM in nanomedicine design has led to the development of biodegradable and biocompatible nanocomposites that reduce potential toxicities and facilitate the targeted release of therapeutic and diagnostic agents based on the patient’s specific tumor characteristics [128]. GBM nanocomposites are promising candidates for personalized anticancer nanomedicine owing to their unique advancements. Surface modification of GBMs enhances biocompatibility and colloidal stability behavior in biological environments. Covalent functionalization with polymers like polyethylene glycol (PEG), poly(vinyl alcohol) (PVA), and chitosan improves pharmacokinetics, reduces non-specific interactions, and facilitates targeted drug delivery. The conjugation of specific ligands (e.g., peptides, folic acid, aptamers, and monoclonal antibodies) to GBMs to enhance drug delivery has shown some success, yet issues like off-target effects and poor tissue penetration still limit their full clinical potential. Beyond drug delivery, the multifunctionality of functionalized graphene materials, such as their ability to enhance PTT and PDT, provides an exciting avenue for future cancer treatments. However, the real-world application of these approaches requires more rigorous clinical validation to ensure their therapeutic efficacy, especially considering the inherent complexities and adaptability of the TME [128,129,130,131].

Personalized GBM nanomedicines can utilize the patient’s unique molecular profile to selectively target specific biomarkers, having the potential to revolutionize cancer therapy. This approach leverages patient-specific tumor characteristics to enhance precision medicine, promising to improve treatment efficacy while minimizing side effects. This innovation in GBM nanomedicine design paves the way for safer and more efficient cancer therapies. For these applications, GBM nanocomposites can be engineered to recognize and bind to specific biomarkers expressed by tumor cells or the surrounding TME, such as overexpressed receptors or abnormal extracellular matrix (ECM) components [44,132,133]. The ability of GBM nanomedicines to target specific biomarkers requires a detailed understanding of the molecular diversity present in the TME. Biomarkers vary considerably across tumor types and even within different regions of the same tumor. While the design of GBMs to bind to these biomarkers may improve precision, the dynamic nature of tumor biology presents a challenge in maintaining long-term effectiveness. For instance, tumor cells can rapidly mutate, leading to the loss or alteration of biomarkers, which can reduce the therapeutic effectiveness of GBMs over time. Incorporating biomarkers into GBMs for personalized therapy involves identifying unique molecular features of a patient’s tumor to tailor treatment. Tumors exhibit diverse genetic, proteomic, and metabolic profiles that influence their behavior and response to therapies. By functionalizing GBM nanomedicines with specific ligands, antibodies, or peptides that target these biomarkers, it is possible to precisely deliver therapeutic agents to tumor cells within the microenvironment. This approach enables the reprogramming of the TME by selectively targeting pro-tumorigenic cells (such as tumor-associated macrophages or fibroblasts) and reprogramming them to an anti-tumor phenotype. This approach shows promise in theory; however, in practice, the ability to efficiently target and modulate these cells in a highly complex and heterogeneous TME is still a significant challenge. Moreover, biomarkers can also guide the design of nanomedicines to overcome resistance mechanisms, such as hypoxia or immune evasion. While this personalized approach enhances the efficiency of treatment, reduces off-target effects, and allows for more effective modulation of the TME, a deeper understanding of the molecular mechanisms driving resistance and immune evasion in the TME is necessary to ultimately improve therapeutic outcomes.

GBM nanocomposites have also emerged as a promising strategy to enhance immune responses in cancer therapy, particularly in overcoming immune evasion mechanisms that hinder the success of current immunotherapies. The ability of GBMs to trigger immune responses, such as activating dendritic cells (DCs) or targeting immune checkpoint inhibitors, holds significant potential for augmenting both innate and adaptive immune responses within the TME [133]. However, while these therapies show promise, they also face several challenges, particularly in their ability to effectively stimulate sustained immune activation in the highly immunosuppressive TME that limits the effectiveness of current treatments. The TME, with its dense extracellular matrix and presence of immunosuppressive cells like regulatory T-cells (Tregs) and myeloid-derived suppressor cells (MDSCs), can limit the effectiveness of even the most advanced immunotherapies. Among GBMs, GO and rGO offer a novel approach to overcoming such limitations by facilitating immune activation to suppress tumorigenesis. GBMs can trigger immunomodulatory effects by interacting with TLRs (Toll-like receptors), stimulating the expression of tumor necrosis factor (TNF)-α and interleukin (IL)-10, IL-1α, and IL-6. Moreover, GBMs can be engaged in macrophage autophagy through the TLR pathways, further inducing immunogenic cell death (ICD), especially through the TLR4 and TLR3 signaling, being essential parts of ICD-stimulated immunogenicity and cytotoxic effects of both CD8+ T cells and NK cells [133,134,135]. However, while initial results are promising, the complexity of immune modulation through these receptors requires more extensive exploration. The signaling pathways triggered by GBMs may not be uniform across all tumor types or immune cell populations, which could lead to variable therapeutic outcomes. Furthermore, while GO and reduced GO (rGO) have been studied as adjuvants in immunotherapy, the long-term stability and effectiveness of these nanomaterials, especially in vivo, remain to be fully assessed.

GBM nanocomposites have also been studied as vaccine carriers, offering a novel approach to enhancing neoantigen-specific immune responses. Traditional vaccines face challenges in sustaining antigen presentation and efficiently stimulating both CD4+ and CD8+ T cell responses, especially with neoantigens derived from tumor mutations. Recent studies by Xu et al. [136] have shown that PEG-rGO functionalized with neoantigens and CpG oligodeoxynucleotide (CpG ODN being a TLR-9 agonist) can act as cancer vaccines, enhancing vaccine delivery and stimulating strong CD8+ T cell responses. PET imaging with 64Cu demonstrated that rGO-PEG significantly enhanced vaccine delivery to lymph nodes (>100-fold compared to soluble vaccines). Additionally, rGO-PEG induced ROS generation in DCs (dendritic cells), leading to endolysosomal alkalization and promoting sustained antigen presentation to T cells. In vivo, a single dose of rGO-PEG vaccination generated strong CD8+ T cell responses and eliminated tumors, making it a promising nanoplatform for personalized cancer vaccination. However, these approaches are not without limitations. For example, the efficiency of antigen presentation and the ability to induce a robust and sustained immune response against neoantigens are still areas of active research. Despite advancements, traditional vaccines face significant hurdles, including sustaining antigen presentation and eliciting efficient immune responses. The challenge remains to ensure that the vaccines not only target tumor cells effectively but also navigate the complex TME, which often includes immune suppression mechanisms that can hinder their efficacy. While research has primarily focused on CD8+ T cell enhancement and antigen presentation [137], there are still many unexplored areas that could further improve GO-based cancer immunotherapy. One particularly promising direction is the modulation of immune evasion pathways, such as the PD-L1/PD-1 axis, which remains an area of great interest for improving the efficacy of GBM-based cancer immunotherapies. While PD-L1/PD-1 inhibitors have shown success in some cancers, their application in combination with GBMs may offer a more comprehensive strategy for overcoming immune suppression in the TME [133]. However, the clinical translation of these promising findings is still hindered by the need for further optimization of GBM formulations, precise targeting strategies, and understanding of their interactions within the immune microenvironment.

In conclusion, while GBMs represent a novel and promising approach to overcoming immune evasion and enhancing cancer immunotherapy, there are still several challenges to address. The need for further research into their immunomodulatory properties, long-term stability, and optimization for personalized treatment regimens remains essential for translating these promising technologies into effective clinical therapies.

## 4. Conclusions

The reprogramming of the TME plays a pivotal role in cancer cell survival, immune evasion, and therapeutic resistance of cancer cells. While cancer cells exploit metabolic shifts, such as the Warburg effect, by transitioning from oxidative phosphorylation to aerobic glycolysis, the subsequent influence on stromal and immune cells contributes significantly to immune suppression and therapy resistance. Despite the promising potential of targeting these metabolic pathways using nanomedicine and advanced therapeutic strategies, the adaptability and complexity of TME metabolism present ongoing challenges. For instance, while graphene-based materials (GBMs) offer promising capabilities in enhancing drug delivery, promoting photothermal and photodynamic therapies, and inducing oxidative stress, their effectiveness remains contingent on overcoming significant barriers such as poor accumulation at the tumor site, potential toxicity, and the dynamic nature of the TME. Furthermore, while metabolic inhibitors targeting glucose and glutamine metabolism are a promising avenue, the heterogeneity and plasticity of cancer cells complicate the consistent targeting of these pathways. Despite these obstacles, recent progress in understanding the TME’s metabolic landscape highlights the need for more comprehensive analyses of tumor metabolomics, oxidative stress mechanisms, and the interplay between the TME and immune responses. Future research should prioritize refining nanomaterials for better targeting, minimizing systemic toxicity, and developing synergistic approaches that combine metabolic modulation with immune checkpoint inhibitors to maximize therapeutic efficacy. Only through a holistic understanding of the TME and continued innovation in treatment strategies can the full potential of these therapies be realized.

## 5. Future Directions

To improve the therapeutic potential of GBMs in TME-targeting cancer treatment, a deeper understanding of mechanisms like oxidative stress, autophagy, and metabolic heterogeneity in cancer cells is critical (Figure 9). While oxidative stress is widely acknowledged as a key factor in cancer progression, its dual role in both promoting and resisting therapy complicates treatment strategies. Oxidative stress refers to the imbalance between the production of ROS and the cell’s ability to neutralize them with the production of antioxidants. These elevated ROS levels can damage cellular components such as DNA, proteins, and lipids, potentially leading to mutations and further promoting cancer cell survival and proliferation. However, cancer cells are also equipped with defense mechanisms to adapt to oxidative stress, such as the upregulation of antioxidant enzymes. Understanding how oxidative stress contributes to tumor survival and resistance to treatments is essential for developing strategies that can either enhance the oxidative stress within tumors or block the cellular defense mechanisms. GBMs have excellent electron transfer capabilities, which can potentially interact or even modulate ROS levels. Thus, the development of GBM nanodrugs can help in the regulation of tumor oxidative stress by either increasing ROS levels to induce cancer cell death or reducing ROS to protect normal cells.

Although GBMs can regulate oxidative stress through localized ROS generation, the challenge lies in selectively increasing ROS levels in tumor cells while protecting normal cells from collateral damage. Despite promising results, research has yet to consistently demonstrate the ability of GBM nanodrugs to effectively target and modulate antioxidant defenses in tumors without causing toxicity to healthy tissues. While GBMs have shown promise in enhancing oxidative stress-induced cancer cell death, cancer cells’ adaptive antioxidant response complicates treatment efficacy. Future research should explore ways to selectively inhibit these antioxidant mechanisms in cancer cells to increase their vulnerability to therapies like chemotherapy and radiation. While GBM-based nanodrugs offer the potential to enhance ROS generation, the challenge remains in overcoming tumor resistance mechanisms that allow cancer cells to persist despite oxidative stress. For GBMs to inhibit the antioxidant defense mechanism in cancer cells, the selective targeting of antioxidant enzymes can be induced, blocking their protective mechanism and rendering tumor cells more susceptible to treatments like chemotherapy or radiation therapy. The combination of GBMs with traditional cancer therapies (chemotherapy, radiation) can create synergistic effects; thus, by increasing oxidative stress, GBM-based nanodrugs could potentially enhance the efficacy of chemotherapy and radiation therapies. The combination of GBMs with traditional therapies shows promise for synergistic effects, but more investigation is needed to optimize dosage, delivery, and timing to maximize efficacy without overwhelming normal tissue. In addition to ROS modulation, GBMs can be functionalized to target multiple pathways involved in tumor survival, such as growth factors, signaling pathways, or drug efflux pumps. Tumors adapt to chronic oxidative stress by activating pathways like NRF2 (Nuclear Factor Erythroid 2-related Factor 2), which upregulates antioxidant proteins. By using GBMs to probe these pathways or to interfere with their activation, the understanding of tumors’ resistance to oxidative damage can be promoted, and new strategies can be developed to reverse this adaptation [138].

The manipulation of autophagy presents another promising, yet complex, strategy for cancer therapy. While autophagy can be protective of cancer cells under stress, its inhibition or promotion may lead to tumor regression. It is a double-edged sword mechanism by which cancer cells degrade and recycle damaged or unnecessary components to survive under nutrient-deprived or stressful conditions, providing them with a survival advantage to resist therapeutics, but one that can also serve as a tumor suppressor mechanism through the disruption of autophagy processes or excessive autophagic flux, further promoting apoptotic cell death. Thus, the role of autophagy in cancer therapy is complex, as it can either support tumor survival or, if manipulated correctly, lead to cell death. Research into autophagy inhibitors or activators is rapidly growing, and future therapeutic approaches may seek to fine-tune this process in cancer cells. Current research indicates that GBMs can function as effective delivery vehicles for autophagy modulators, either inducing cell death or supporting survival depending on the manipulation of autophagic pathways. For example, autophagy-inducing agents can be delivered by GBMs to activate autophagy, potentially forcing cancer cells to undergo autophagic cell death. By specifically functionalizing GBMs to target autophagic pathways, they could assist either in inhibiting cancer cells’ survival or in promoting cell death. Such functionalization could be followed by peptides or agonists binding to autophagy proteins such as Beclin-1 agonist, mTOR inhibitors such as rapamycin, ligands for targeting autophagy-related receptors such as LC3 (microtubule-associated light chain 3 involved in autophagosome formation) or monoclonal antibodies specific to autophagy-regulating proteins or receptors like Beclin-1, LC3, or PI3K (phosphoinositide 3 kinase) that could improve the selectivity of the GBMs. The interaction of GBMs with autophagy signaling pathways could affect key autophagy-regulating proteins, such as Beclin-1, LC3, mTOR, and AMPK (AMP-activated protein kinase), either enhancing or suppressing autophagy; thus, modulating the survival or death of cancer cells. However, much of the existing data is inconsistent, and there is a need for a more comprehensive understanding of how GBMs can be tuned to target specific autophagic proteins, such as Beclin-1, LC3, or mTOR. The next step is to identify the most effective combinations of GBMs and autophagy regulators that can effectively manage autophagy regulation. In cancer cells, autophagy often serves as a survival mechanism under oxidative stress. By using GBMs to stimulate oxidative stress, it may be possible to overwhelm cancer cells, promoting a shift from autophagic survival to autophagic cell death. By either inhibiting or activating autophagy, GBMs could enhance the efficacy of chemotherapy and radiation therapy by preventing cancer cells from repairing the damage induced by the treatments. Therefore, autophagy modulation could be integrated with GBM as a valuable strategy in overcoming therapy resistance [139].

These factors play a critical role in tumor progression and drug resistance, making them key areas of focus for improving GBM-based therapeutic strategies that are known for their ability to induce localized oxidative stress. Additionally, future research should aim to overcome the metabolic flexibility of cancer cells, which allows them to adapt and survive under stress conditions. Cancer cells have developed the ability to adapt to various stress conditions, such as nutrient deprivation and oxidative stress, by reprogramming their metabolism. By exploring the ways in which these mechanisms influence cancer cell behavior, researchers can develop more targeted and effective treatment strategies that harness the unique properties of GBMs. Integrating multi-modal therapeutic approaches, which combine GBMs with other treatments including PTT/PDT, ultrasound therapy, and magnetic targeting, could provide a more direct strategy for tackling cancer’s metabolic plasticity. Moreover, GBMs can be used to deliver drugs directly to tumors, disrupt the tumor microenvironment, or even induce localized oxidative stress. When combined with other treatments, such as targeted therapies, chemotherapy, or immunotherapy, these materials could enhance the overall therapeutic efficacy by attacking multiple vulnerabilities in cancer cells. A multi-pronged approach would not only help tackle the tumor’s metabolic plasticity but also prevent resistance from developing, offering a more comprehensive and potent strategy against cancer. While GBMs are capable of inducing localized oxidative stress, tumor cells may counteract this by activating compensatory metabolic pathways like NRF2. Future studies should focus on combining GBMs with metabolic inhibitors to disrupt these pathways, limiting the cells’ ability to adapt and potentially enhancing treatment outcomes. Therapies designed to inhibit key metabolic enzymes or pathways could potentially limit the ability of cancer cells to adapt to this environment. Although GBMs can be functionalized to target these pathways, a critical question remains: how can we achieve consistent and selective targeting of metabolic vulnerabilities without promoting compensatory mechanisms that reduce treatment efficacy? Future research on identifying specific metabolic vulnerabilities and designing therapeutic approaches to deprive the cellular energy sources, rendering them more susceptible to treatment, could open the way for more effective therapies [140].

Finally, the integration of personalized treatment strategies based on the metabolic profiles of individual tumors holds great potential. Tumor metabolomics and proteomics offer the opportunity to tailor therapies to the unique characteristics of each cancer. Refining the analysis of tumor metabolomes will offer a direct understanding of the metabolic profiles specific to different cancer types, thereby enhancing the precision and effectiveness of metabolic-targeted therapies. The complexity of cancer metabolism has long been an obstacle to developing highly effective therapies. The unique metabolic profiles exhibited by distinct cancer types, in combination with metabolic alterations within the same type, demonstrate significant metabolic heterogeneity. By identifying unique metabolic signatures, a more precise understanding could unlock new potentials for metabolic-targeted therapies. GBMs show promise in being integrated into these personalized therapeutic regimens, with the potential for targeting specific metabolic targets identified through metabolomic profiling. The integration of advanced technologies like metabolomics and proteomics, combined with GBMs, could facilitate the identification of key metabolic targets and enable the design of highly personalized therapeutic regimens. The properties of GBMs align perfectly with the increased level of specificity and effectiveness required for metabolic-targeted therapies. Moreover, GBMs can interact with the TME, inducing oxidative stress and modulating the metabolic pathways of cancer cells, and enhancing the therapeutic effect, as presented in this review. The combination of tumor metabolomics with GBM-targeted therapies could be crucial for the development of precise and more effective personalized therapies, improving cancer treatment outcomes [141]. However, despite the advances in metabolomics, much work remains to be done to define the metabolic vulnerabilities that are specific to tumor subtypes. The development of personalized therapies will be contingent upon accurately identifying these vulnerabilities and ensuring that GBM-based treatments are capable of selectively targeting them.

In summary, the integration of GBMs with therapies targeting oxidative stress, autophagy, and metabolic reprogramming represents a promising approach in cancer treatment. However, critical challenges persist, including the inconsistent response of tumors to oxidative stress modulation, the complexity of autophagy regulation, and the metabolic adaptability of cancer cells. Future research should address these challenges by focusing on selective targeting and combination therapies that can overcome the existing resistance mechanisms, ultimately advancing GBM-based therapies into more effective and personalized cancer treatment options.

## Figures and Tables

**Figure 1 ijms-26-04525-f001:**
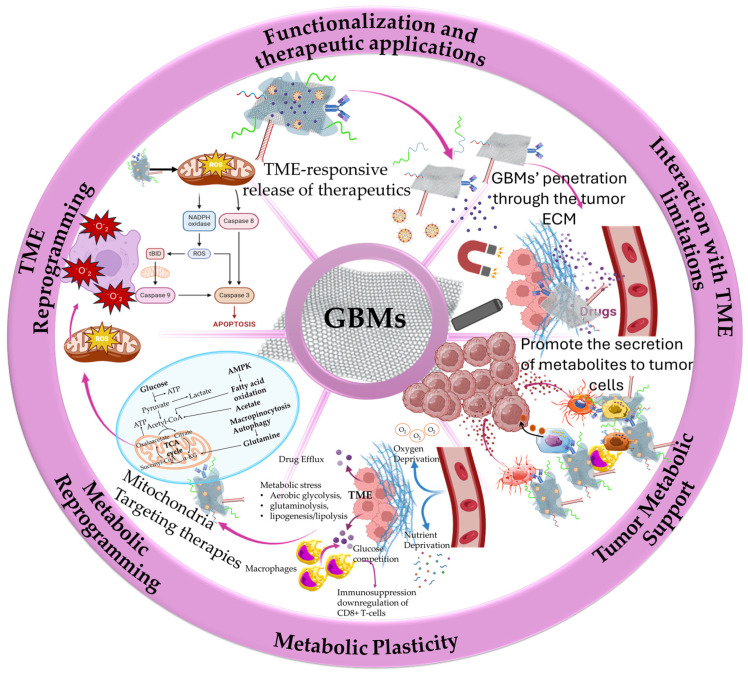
Representation of the applications of GBMs in TME reprogramming through mitochondria targeting. Color explanation of cells in the scheme: Orange-blue, pink, yellow-purple, blue-purple represent various types of immune cells, such as T cells and dendritic cells (Created in BioRender.com by Athina Angelopoulou (2025) “https://app.biorender.com/user/signin?illustrationId=6156d45891063d00af8af51d” (accessed on 18–23 March 2025) and Microsoft ppt).

**Figure 2 ijms-26-04525-f002:**
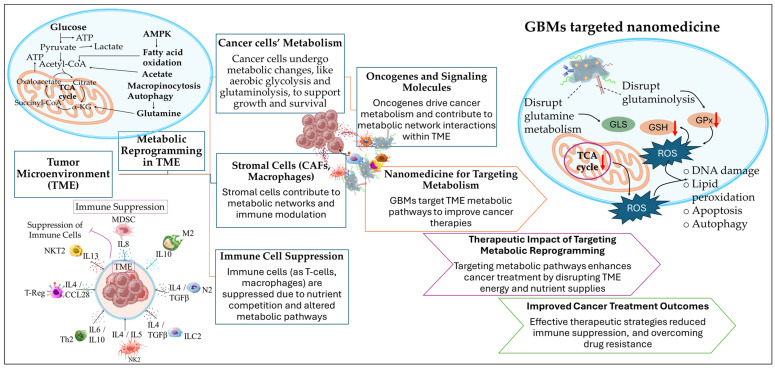
Schematic representation of the application of GBM nanocomposites in the metabolic reprogramming in the TME and the advantages of nanomedicine in targeting tumor metabolism (Created in BioRender.com by Athina Angelopoulou (2025) “https://app.biorender.com/user/signin?illustrationId=6156d45891063d00af8af51d” (accessed on 18–23 March 2025) and Microsoft ppt).

**Figure 3 ijms-26-04525-f003:**
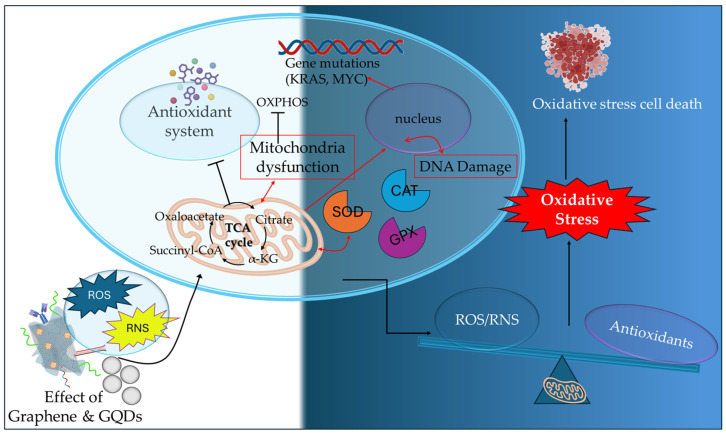
Schematic representation of the effect of Gr and GQD nanocomposites on the antioxidant defense of mitochondria to promote ROS imbalance and oxidative-stress-triggered cell death (Created in BioRender.com by Athina Angelopoulou (2025) “https://app.biorender.com/user/signin?illustrationId=6156d45891063d00af8af51d” (accessed on 18–23 March 2025) and Microsoft ppt).

**Figure 4 ijms-26-04525-f004:**
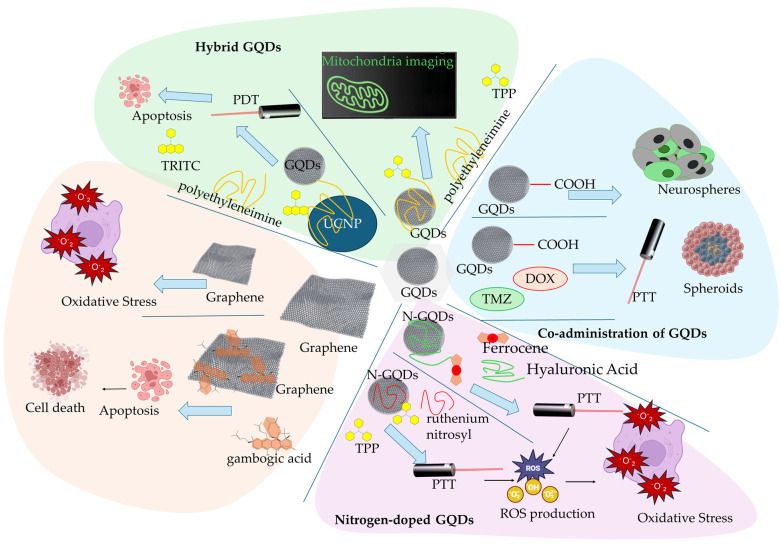
Analytical schematic illustration of the different examples of graphene (Gr) and graphene quantum dots (GQDs)-based nanocomposites in cancer therapeutics (Created in BioRender.com by Athina Angelopoulou (2025) “https://app.biorender.com/user/signin?illustrationId=6156d45891063d00af8af51d” (accessed on 23–30 April 2025) and Microsoft ppt).

**Figure 5 ijms-26-04525-f005:**
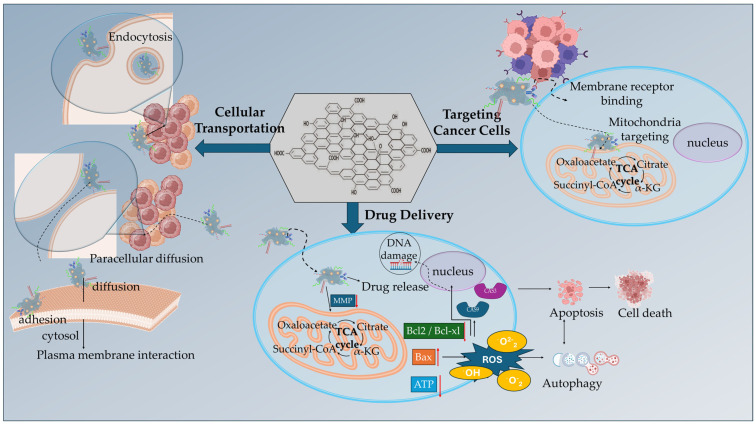
Visual representation of the role of GO nanocomposites in targeting cancer cells and mitochondria for enhanced drug delivery and effective stimulation of autophagy and apoptotic cell death through ROS production. Explanation of red arrows: pointing upwards = upregulation, pointing downwards = downregulation (Created in BioRender.com by Athina Angelopoulou (2025) “https://app.biorender.com/user/signin?illustrationId=6156d45891063d00af8af51d” (accessed on 18–23 March 2025) and Microsoft ppt).

**Figure 6 ijms-26-04525-f006:**
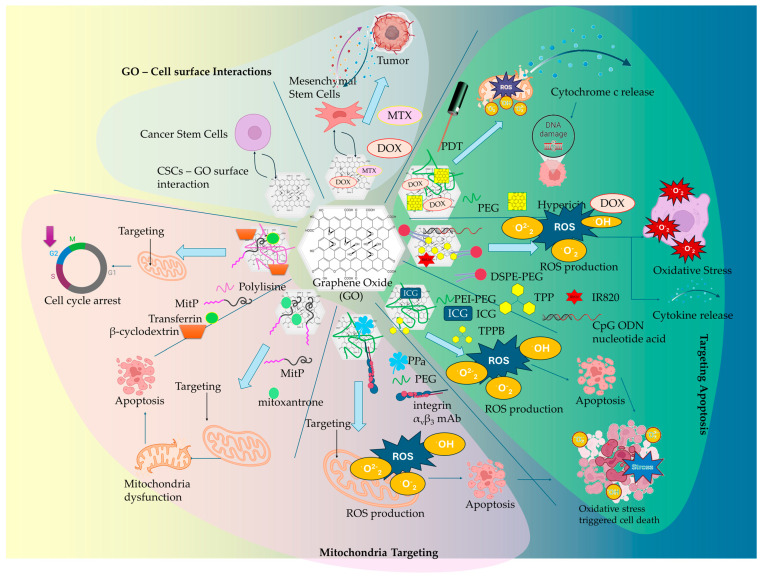
An analytical schematic illustration of the different examples of graphene oxide (GO)-based nanocomposites in cancer therapeutics (Created in BioRender.com by Athina Angelopoulou (2025) “https://app.biorender.com/user/signin?illustrationId=6156d45891063d00af8af51d” (accessed on 23–30 April 2025) and Microsoft ppt).

**Figure 7 ijms-26-04525-f007:**
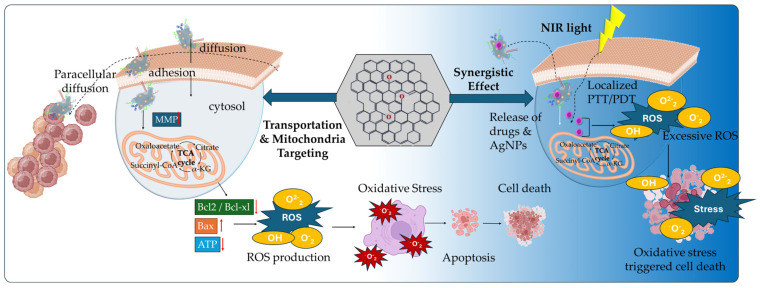
Schematic of the role of rGO nanocomposites in targeting cancer cells and mitochondria for enhanced drug delivery to promote apoptotic cell death and synergistic effect with AgNPs to further promote oxidative stress and trigger cancer cell death. Explanation of red arrows: pointing upwards = upregulation, pointing downwards = downregulation (Created in BioRender.com by Athina Angelopoulou (2025) “https://app.biorender.com/user/signin?illustrationId=6156d45891063d00af8af51d” (accessed on 18–23 March 2025) and Microsoft ppt).

**Figure 8 ijms-26-04525-f008:**
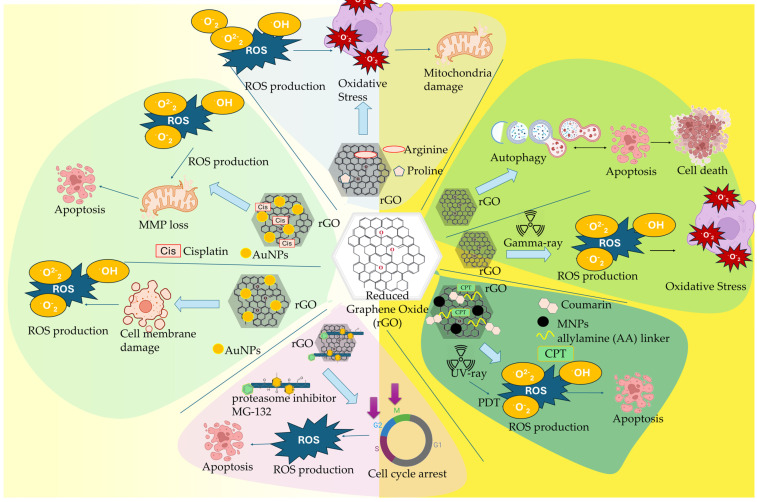
A schematic illustration of varied examples of reduced graphene oxide (rGO)-based nanocomposites in cancer therapeutics (Created in BioRender.com by Athina Angelopoulou (2025) “https://app.biorender.com/user/signin?illustrationId=6156d45891063d00af8af51d” (accessed on 23–30 April 2025 and Microsoft ppt).

**Figure 9 ijms-26-04525-f009:**
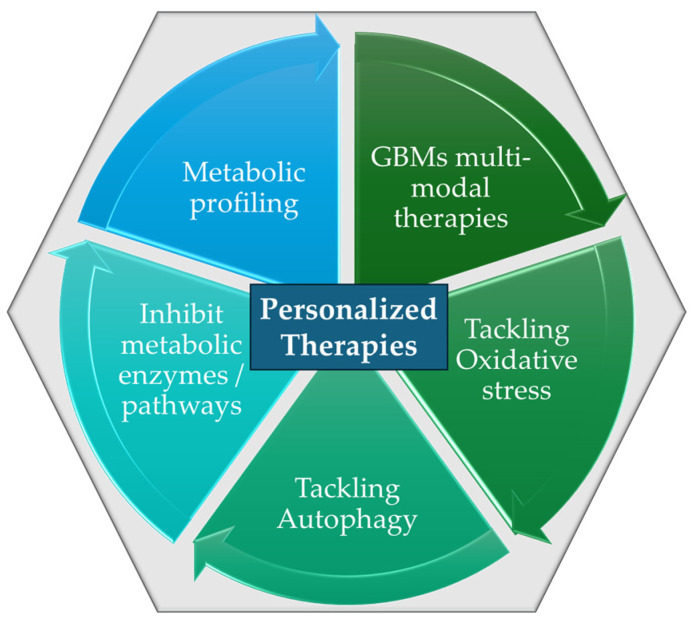
Schematic illustration of the future directions on GBMs.

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
