# Peer review of "Graphene Nanocomposites in the Targeting Tumor Microenvironment: Recent Advances in TME Reprogramming"

_ijms, 2025, doi:10.3390/ijms26104525_

Round 1
Reviewer 1 Report
Comments and Suggestions for Authors
The review by Angelopoulou A. and coworker, titled "Graphene Nanocomposites in Targeting Tumor Microenvironment:Advances in TME Reprogramming" presents a high-quality and well-structured manuscript. The topic is of great relevance and interest within the field of nanomedicine and oncology, and the authors have successfully compiled up-to-date and cutting-edge information that provides valuable insight into the use of graphene-based nanocomposites in modulating the tumor microenvironment.
Dear editors
The review by Angelopoulou A. and coworker, titled "Graphene Nanocomposites in Targeting Tumor Microenvironment:Advances in TME Reprogramming" presents a high-quality and well-structured manuscript. The topic is of great relevance and interest within the field of nanomedicine and oncology, and the authors have successfully compiled up-to-date and cutting-edge information that provides valuable insight into the use of graphene-based nanocomposites in modulating the tumor microenvironment.
However, while the content is scientifically sound and informative, the manuscript would greatly benefit from a more in-depth discussion of the different types of nanocomposites formed between graphene and various materials such as PEG, etc. In particular, it is recommended that the authors provide specific examples of the main types of graphene-based nanocomposites used in this context, ideally through schematic illustrations and clear drawings where these composites appear. These visual aids would complement the information presented in the tables and enhance readers' overall understanding.
Additionally, it is important to note that in several of the figures included in the manuscript, the text and labeling are extremely difficult to read due to the small font size. Increasing the legibility of these figures by using larger fonts or redesigning them for clarity would significantly improve the readability and overall quality of the review.
Incorporating these suggestions would strengthen the impact and clarity of the manuscript and help the reader fully understand the potential and complexity of graphene nanocomposites in targeting the tumor microenvironment.
Additionally, it is important to note that in several of the figures included in the manuscript, the text and labeling are extremely difficult to read due to the small font size. Increasing the legibility of these figures by using larger fonts or redesigning them for clarity would significantly improve the readability and overall quality of the review.
Incorporating these suggestions would strengthen the impact and clarity of the manuscript and help the reader fully understand the potential and complexity of graphene nanocomposites in targeting the tumor microenvironment.
Author Response
Dear Reviewer,
Thank you for your supportive comments on the Review article we have submitted.
Here we present our responses.
Reviewer Remarks: Dear editors
The review by Angelopoulou A. and coworker, titled "Graphene Nanocomposites in Targeting Tumor Microenvironment:Advances in TME Reprogramming" presents a high-quality and well-structured manuscript. The topic is of great relevance and interest within the field of nanomedicine and oncology, and the authors have successfully compiled up-to-date and cutting-edge information that provides valuable insight into the use of graphene-based nanocomposites in modulating the tumor microenvironment.
Incorporating these suggestions would strengthen the impact and clarity of the manuscript and help the reader fully understand the potential and complexity of graphene nanocomposites in targeting the tumor microenvironment.
Comment 1: However, while the content is scientifically sound and informative, the manuscript would greatly benefit from a more in-depth discussion of the different types of nanocomposites formed between graphene and various materials such as PEG, etc. In particular, it is recommended that the authors provide specific examples of the main types of graphene-based nanocomposites used in this context, ideally through schematic illustrations and clear drawings where these composites appear. These visual aids would complement the information presented in the tables and enhance readers' overall understanding.
Response 1: We sincerely thank the reviewer for the constructive suggestion. In response, we have expanded our discussion on graphene-based nanocomposites throughout the article. To maintain the critical perspective and scientific focus of our review article, we have presented these examples in a more analytical and evaluative manner, rather than simply listing different combinations. Following the reviewers’ suggestions, the incorporated examples have been presented in an approach that aligns with the overall scope and objectives of our manuscript (Advances in TME Reprogramming), offering readers a deeper and more meaningful understanding of the field. Relevant modifications have been highlighted in yellow color. To further enhance clarity and reader comprehension, we have also added schematic illustrations and detailed drawings that visually depict the structures of these composites, complementing the information already provided in the tables. We believe these additions significantly improve the depth and overall quality of the manuscript, as suggested.
Relevant changes can be found in Section [2.1, 2.2, 2.3 ] and Figures 4, 6, and 8.
Comment 2: Additionally, it is important to note that in several of the figures included in the manuscript, the text and labeling are extremely difficult to read due to the small font size. Increasing the legibility of these figures by using larger fonts or redesigning them for clarity would significantly improve the readability and overall quality of the review.
Response 2: We thank the reviewer for highlighting this important point. In response, we have carefully revised the figures throughout the manuscript by increasing the font size of the text and labeling to ensure better readability and clarity. All updated figures can be found in the revised version of the manuscript.
Reviewer 2 Report
Comments and Suggestions for Authors
The manuscript by Kolokithas-Ntoukas et al. discusses research studies on GBM nanocomposites aimed at enhancing biodegradability, minimizing toxicity, and improving the efficacy of therapeutic agent delivery, all with the goal of reprogramming the tumor microenvironment for effective anticancer therapy.
I have a few suggestions for polishing the manuscript.
- The presentation of a collection of previous studies and their results holds some interest; however, a critical assessment of the available research would provide far greater value. The critical analysis ought to maintain a coherent structure, for instance: What aspects have been executed effectively? What has been executed inadequately? What tasks are left to complete and what is the best approach to accomplish them? Authors are expected to present their perspectives on the prior reports using precise and analytical terminology.
- The authors should add a paragraph highlighting how graphene oxide targets tumor microenvironment through interaction with the cancer stem cells.
- In the Introduction Section the authors should add more references in the paragraph stated “ Graphene (Gr) and graphene-based materials (GBMs), such as graphene oxide (GO) and reduced graphene oxide (rGO) have gained significant attention in ……biological properties”. The authors should review the following articles such as
https://doi.org/10.36922/gtm.4602
https://doi.org/10.1016/j.msec.2019.109774
Author Response
Dear Reviewer,
Thank you for your supportive comments on the Review article we have submitted.
Here we present our responses.
Reviewer Remarks: The manuscript by Kolokithas-Ntoukas et al. discusses research studies on GBM nanocomposites aimed at enhancing biodegradability, minimizing toxicity, and improving the efficacy of therapeutic agent delivery, all with the goal of reprogramming the tumor microenvironment for effective anticancer therapy.
I have a few suggestions for polishing the manuscript.
Comment 1: The presentation of a collection of previous studies and their results holds some interest; however, a critical assessment of the available research would provide far greater value. The critical analysis ought to maintain a coherent structure, for instance: What aspects have been executed effectively? What has been executed inadequately? What tasks are left to complete and what is the best approach to accomplish them? Authors are expected to present their perspectives on the prior reports using precise and analytical terminology.
Response 1: We thank the reviewer for this valuable comment. In response, we have revised the review article and especially part 2 (including 2.1, 2.2 and 2.3) and part 5 (Future Directions) of the manuscript to include a more critical and structured analysis of the existing literature. Specifically, we discussed their strengths and weaknesses and highlighted the current gaps in knowledge. Furthermore, we have added a dedicated section presenting our perspectives on the most promising future directions for the application of graphene-based nanomaterials in targeting the tumor microenvironment.
Comment 2: The authors should add a paragraph highlighting how graphene oxide targets tumor microenvironment through interaction with the cancer stem cells.
Response 2: Thank you for your valuable suggestion. We have taken your comment into consideration and added a dedicated paragraph in Section 2.2, where we discuss in detail how graphene oxide (GO) targets the tumor microenvironment through its interactions with cancer stem cells (CSCs). This addition highlights the potential mechanisms by which GO selectively affects CSCs, including the inhibition of key signaling pathways such as Wnt, Notch, and STAT, thereby contributing to the differentiation and eradication of CSCs within tumors. We believe this addition enhances the clarity of our work in addressing the role of GO in targeting the tumor microenvironment.
Comment 3: In the Introduction Section the authors should add more references in the paragraph stated “ Graphene (Gr) and graphene-based materials (GBMs), such as graphene oxide (GO) and reduced graphene oxide (rGO) have gained significant attention in ……biological properties”. The authors should review the following articles such as
https://doi.org/10.36922/gtm.4602
https://doi.org/10.1016/j.msec.2019.109774
Response 3: We would like to thank the reviewer for this valuable suggestion. In the revised manuscript, we have enriched the Introduction section. Specifically, we have incorporated the suggested articles, highlighting the unique physicochemical properties, high drug-loading capabilities, fluorescence imaging stability, and enhanced photothermal and cytotoxic activities of graphene and its derivatives. These additions aim to strengthen the scientific background and improve the completeness of the Introduction. The related modifications can be found on page [1-2], lines [36-42].
Round 2
Reviewer 2 Report
Comments and Suggestions for Authors
The manuscript by Kolokithas-Ntoukas et al. discusses research studies on GBM nanocomposites aimed at enhancing biodegradability, minimizing toxicity, and improving the efficacy of therapeutic agent delivery, all with the goal of reprogramming the tumor microenvironment for effective anticancer therapy.
The authors have addressed all the previous comments. Thus, the manuscript can be accepted in its present form.